# The essential elements for the noncovalent association of two DNA ends during NHEJ synapsis

Bailin Zhao [1], Go Watanabe [1], Michael J. Morten[2], Dylan A. Reid[2], Eli Rothenberg[2] & Michael R. Lieber[1]

One of the most central questions about the repair of a double-strand DNA break (DSB) concerns how the two free DNA ends are brought together — a step called synapsis. Using single-molecule FRET (smFRET), we show here that both Ku plus XRCC4:DNA ligase IV are necessary and sufficient to achieve a flexible synapsis of blunt DNA ends, whereas either alone is not. Addition of XLF causes a transition to a close synaptic state, and maximum efficiency of close synapsis is achieved within 20 min. The promotion of close synapsis by XLF indicates a role that is independent of a filament structure, with action focused at the very ends of each duplex. DNA-PKcs is not required for the formation of either the flexible or close synaptic states. This model explains in biochemical terms the evolutionarily central synaptic role of Ku, X4L4, and XLF in NHEJ for all eukaryotes.

[1] Department of Pathology, Department of Biochemistry & Molecular Biology, Department of Molecular Microbiology & Immunology, and Section of Computational & Molecular Biology, USC Norris Comprehensive Cancer Center, University of Southern California Keck School of Medicine, Los Angeles, CA 90089, USA. [2] Department of Biochemistry and Molecular Pharmacology, New York University School of Medicine, New York, NY 10016, USA. Correspondence and requests for materials should be addressed to E.R. (email: eli.rothenberg@nyumc.org) or to M.R.L. (email: lieber@usc.edu)

NHEJ is the dominant double-strand break (DSB) repair pathway during most of the cell cycle and must have arisen as the major DNA repair pathway for DSBs early during evolution[1,2]. This may explain the universal presence of Ku in all eukaryotic organisms and the presence of homologs in prokaryotes[3].

One of the most intriguing and important questions in this major DNA repair pathway concerns how the two DNA ends at a DSB are brought together into physical proximity — a process called synapsis. Disparate conclusions have been drawn about this step using data gathered by various methods, and none of them are entirely consistent with genetic observations. Pull-down assays using either purified proteins or cell-free extracts have variously reported that Ku alone[4], that DNA-PKcs alone[5], or that DNA-PK holoenzyme (formed by Ku plus DNA-PKcs)[6] could mediate synapsis of dsDNA. It has been further reported that XRCC4:DNA ligase IV (X4L4) can stabilize the synaptic complex by DNA-PK[7]. Using single-molecule nanomanipulation and purified proteins, it was reported that DNA-PK holoenzyme was the minimal unit for end synapsis on the 100 ms timescale, and addition of PAXX and/or XLF plus X4L4 could further stabilize the synaptic complex to seconds, or even to the minute time-scale[8]. Based on the timescale of synaptic complexes, Wang et al. proposed a stepwise assembly model for NHEJ synaptic machinery. In their experimental system, two dsDNA molecules are tethered together using a dsDNA bridge, to which NHEJ proteins such as XLF and X4 can also bind. The dsDNA bridge and protein binding at nonphysiologic locations on the experimental scaffold might influence synapsis. Importantly, synapsis efficiency was similar when any single protein other than Ku or DNA-PKcs was omitted from the reaction, and this is inconsistent with the stepwise synapsis model that they proposed. Moreover, no ligation was observed in their system when DNA-PKcs was omitted, which is contrary to previous results from ensemble [biochemical (bulk solution) system] studies that indicate DNA-PKcs is not necessary for ligation[9,10].

In another study that utilized the smFRET method and crude extracts, a two-stage end synaptic process was described, proposing that Ku and DNA-PKcs were required for the long-range, indirect interaction of two blunt DNA ends[11]. Also, XLF, non-catalytic X4L4 activity, and DNA-PKcs kinase activity were required for transition to a short-range synapsis in their system. Other components in the *Xenopus laevis* egg extracts beyond just the NHEJ proteins might competitively bind to the dsDNA end, which could affect the synapsis process mediated by NHEJ factors. Importantly, initial smFRET and ensemble ligation studies with purified proteins have found that DNA-PKcs is not necessarily required either for synapsis or for covalent ligation[9,12,13]. Finally, signal joint formation during V(D)J recombination clearly does not require DNA-PKcs, indicating that it is not required for synapsis during NHEJ[14,15].

Ensemble biochemical assays using purified Ku and X4L4 showed that these proteins could mediate efficient ligation of blunt end dsDNA, but such a system cannot provide insight into the synapsis step prior to ligation[9]. Inside the nucleus, the two DNA ends arising from a single dsDNA break can and do diffuse apart[16]. This means that the primary factor in the entire repair process involves bringing those two DNA ends back into proximity to permit repair.

Here, with smFRET, we can directly study this critical synapsis step. We find that, while Ku alone is insufficient for synapsis, Ku plus X4L4 do bring blunt end DNA termini into a 'flexible synapsis' (**FS**). Addition of XLF increases the stability of the synaptic complex and promotes the end-to-end alignment of the dsDNA ends into a 'close synapsis' (**CS**) that is ready for ligation. Addition of DNA-PKcs does not have a significant effect on synapsis; and this may not be surprising, given that DNA-PKcs appeared on the evolutionary stage after the invertebrate-to-vertebrate transition, and thus hundreds of millions of years after the inception of NHEJ. The duration of synapsis, the transitions between discernable FRET states, the population accumulation of synaptic complexes, the time dependence with which synaptic complexes form are all key elements which are discernable using purified proteins, and thus provide information here that was previously unclear. Based on these, we are able to formulate the clearest understanding of the essential elements of the NHEJ synaptic complex.

## Results

**Experimental system for NHEJ synapsis.** Our system relies on fully purified human NHEJ proteins: Ku, X4L4, XLF, PAXX, and DNA-PKcs (Supplementary Fig. 1a,c), as well as defined duplex DNA molecules (lengths 74–85 bp). One duplex DNA is immobilized on the slide surface via a biotin-neutravidin-biotin strategy. The other DNA duplex is free in solution and introduced into the reaction chamber together with NHEJ proteins (Fig. 1a). We find equivalent results when we use L4 that has a C-terminal His tag or when it has an additional N-terminal SNAP tag (Supplementary Fig. 1b), though the latter is used for most of the studies here. The duplex DNA molecules are labeled with Cy3 (donor) or Cy5 (acceptor) located 4 bp from the terminus via the phosphodiester bonds in the backbone. We monitor the smFRET signals for Cy3–Cy5 interaction over short- and long-term time courses. As no phosphate group exists at the 5′ end of either duplex, we can observe the synapsis process without complications due to actual covalent ligation. To simplify the observation, a loop structure is located at the distal end of the incoming duplex in order to monitor synapsis at the blunt DNA end, which is cyanine dye (Cy3 or Cy5) labeled 4 bp from the end. The surface immobilized DNA is labeled with Cy5 and the incoming duplex from the solution has Cy3 near the end; therefore, no FRET signal is detected unless the two duplexes form a synaptic complex. The simultaneous appearance of Cy3 and Cy5 signals documents the interaction of the two duplex DNA ends, and any changes of FRET signal after initial synapsis also can be detected. Therefore, we can monitor the synapsis close to the dsDNA ends in real time as well as any intermediates that develop after initial synapsis. The FRET value ($E_{FRET}$) of standard FRET pairs shows a dependence on the distance between the two dyes (Supplementary Fig. 1d), which indicates that our system is effective for $E_{FRET}$ measurement.

**Ku plus X4L4 can mediate synapsis.** We first tested the contribution of Ku and X4L4 to the synapsis. The efficiency of synapsis is obtained from colocalized Cy3 and Cy5 pairs detected after injection of Cy3-labeled duplex together with NHEJ factors. The clear appearance of both donor and acceptor signals after introduction of active X4L4 together with Ku protein (Fig. 1b) indicates X4L4 can stimulate synapsis between blunt end DNA duplexes. The synapsis efficiency resulting from X4L4 and Ku confirms the stimulation by X4L4 [compare the efficiency from X4L4 and Ku (Supplementary Fig. 1e, lane 1) with that from Ku alone (Supplementary Fig. 1e, lane 2)]. Compared with the synapsis mediated by Ku plus X4L4 (Supplementary Fig. 1e, lane 1), synapsis resulting from Ku alone (lane 2) is negligible, which suggests Ku cannot mediate efficient synapsis. Dwell time histograms were generated based on the synaptic complex observations. This only includes the synapsis events with both beginning and end times within the detection time window (Fig. 1c, left panel). Results mediated by Ku and X4L4 show an interaction of 14.3 (±0.6) s (Fig. 1c), which is relatively long at the molecular

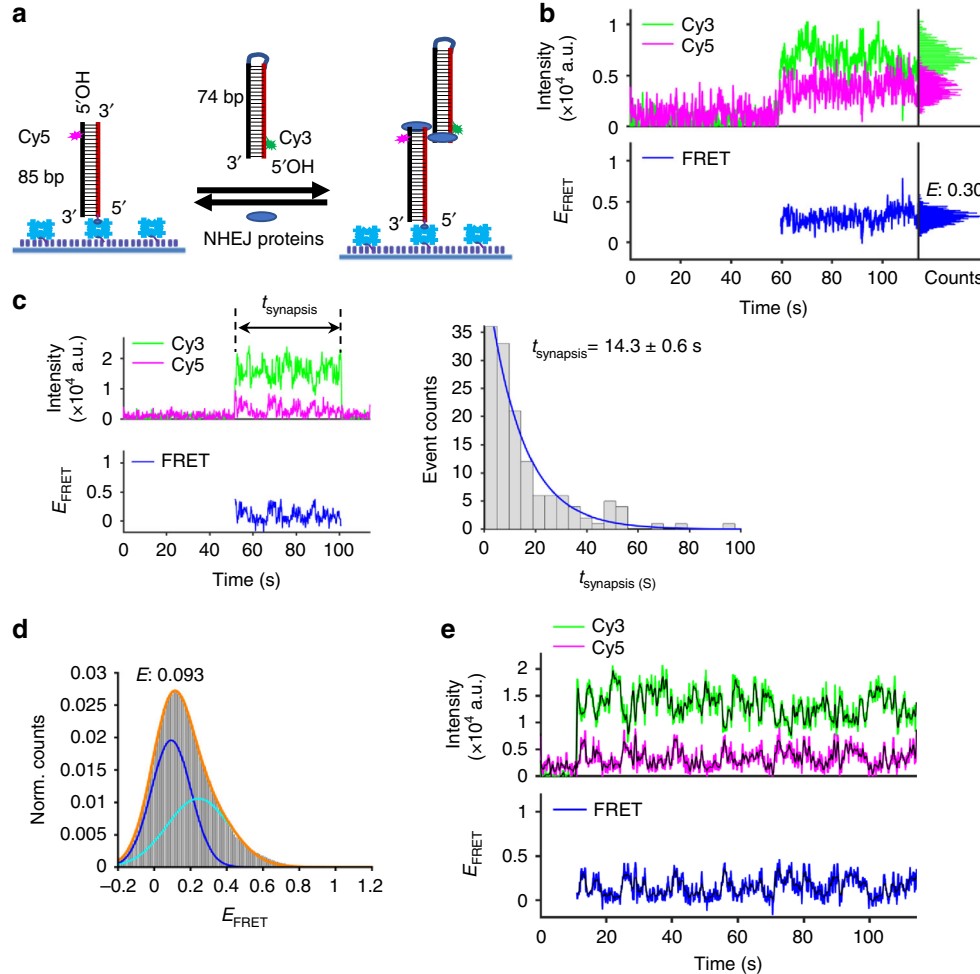

**Fig. 1** Ku and X4L4 can mediate efficient noncovalent synapsis of blunt dsDNA ends. **a** Schematic of the smFRET assay for blunt end synapsis. An 85 bp duplex with a terminal 5′-OH and a Cy5 dye located 4 bp away from the end was immobilized on the surface of the imaging coverslip. The 74 bp incoming dsDNA with a terminal 5′-OH and a Cy3 dye located 4 bp away from the end was added to the reaction via an injected solution that also contained 25 nM Ku and 50 nM X4L4. The figure illustrates a lateral configuration that may permit movement of the duplexes along one another [rather than an end-to-end configuration (see below)]. **b** Representative single-molecule time traces of donor (green) intensity, acceptor (magenta) intensity, and the corresponding $E_{FRET}$ values (blue) for synapsis mediated by Ku and X4L4. Histograms of donor intensity (green), acceptor intensity (magenta), and $E_{FRET}$ values (blue) within synapsis period are indicated at the right part of the figure. **c** Left panel: representative trace for dwell time ($t_{synapsis}$) calculation. Right panel: histogram and corresponding exponential fit of total synapsis time mediated by Ku and X4L4. Cy3 signal lifetime (including zero-FRET and detectable FRET portions) of the synaptic complex was used to calculate the total dwell time for each synapsis event and only the synapsis events ($n = 139$) with both start and end time points within the detection time window were included. Synapsis time shown on graph is represented as mean ± SD of two replicates. **d** Histogram of $E_{FRET}$ values of all synapsis events mediated by Ku and X4L4. The $E$ value shown on the graph was obtained from a Gaussian fit of the highest peak. $n = 515$ molecules. **e** One of the dynamic intensity traces (donor: green; acceptor: magenta) and corresponding $E_{FRET}$ values (blue) of synapsis mediated by Ku and X4L4. The black line represents the smoothed trace of corresponding donor, acceptor, or $E_{FRET}$ trace. Source data are provided as a Source Data1 file

level. This dwell time means Ku and X4L4 can mediate a durable synapsis of two blunt duplexes.

**Ku and X4L4 bring the duplexes into a lateral configuration.** We further examined the relative position of two duplex ends by checking the $E_{FRET}$ profile of a single pair of molecules and also the $E_{FRET}$ distribution of all the molecules in a given experiment. The $E_{FRET}$ values are <0.5 for most of the time points, as shown for the representative time trace (Fig. 1b). The $E_{FRET}$ distribution peak of this molecule is ~0.3, and the $E_{FRET}$ distribution of all the molecules exhibits a main peak of ~0.09 (Fig. 1d), which is much smaller than that obtained from the pre-ligated positive control under the same buffer condition (E: ~0.8 in Supplementary Fig. 4f). Also, 97% of synapsis events have an $E_{FRET}$ <0.6 (Fig.1d

and Supplementary Fig. 1f), which represents the lower boundary of the $E_{FRET}$ distribution of the pre-ligated positive control (Supplementary Fig. 4f). These results show that the two duplex ends are brought together by Ku and X4L4, but that a nanometer-scale distance still exists between these two ends within the synaptic complex.

One possible location of interaction that can result in this low $E_{FRET}$ synaptic complex is in the midsection of each duplex (Fig. 1a). That is, the two duplexes might contact each other in the presence of Ku and X4L4, but at a distant position from the labeled ends, namely in a lateral alignment (i.e., side-by-side and parallel) of the two dsDNA molecules in such a way that the two duplexes could ratchet along one another. We tested the possibility of lateral alignment using DNA duplexes with Cy5 located in the middle of the surface immobilized substrate (40 bp

away from the DNA end) and Cy3 at the terminus (4 bp away from the end) of the substrate that is added in solution (Supplementary Fig. 1g). The number of detected nonzero FRET molecules (which contain at least one nonzero FRET portion within the FRET trajectory) for the probe with midpoint labeling is 64% of that observed with end labeling (Supplementary Fig. 1g), which indicates that the duplexes can make contact in the midsection of each other within the synaptic complex formed by Ku and X4L4. The $E_{FRET}$ distribution of all identified molecules from the midpoint-labeled configuration exhibits a broad distribution (Supplementary Fig. 1h), which confirms the mid-region binding of each duplex. These results above suggest that Ku and X4L4 mediate synapsis by bringing the two duplexes into a parallel lateral alignment. However, a greater number of detected nonzero FRET molecules is observed for the end-labeled configuration compared with the midpoint-labeled duplex (Supplementary Fig. 1g), suggesting that the binding of the two duplexes occurs preferentially near the DNA ends.

The $E_{FRET}$ of some of the synaptic complexes is not stable but is dynamic over time (Fig. 1e), where at some time points, the $E_{FRET}$ value reaches zero (no FRET interaction). This means that the two duplex ends brought together by Ku and X4L4 can move towards and away from each other as the two duplexes remain in a lateral alignment. Since the two duplexes are simply brought into a lateral configuration [rather than end-to-end (see below)] by Ku and X4L4, and the two DNA duplexes can move along one another, we refer to this state as flexible synapsis (**FS**)(Fig. 1a).

**DNA-PKcs has little effect on the FS mediated by Ku and X4L4.** We then tested the effect of DNA-PKcs on the flexible synapsis observed above by Ku and X4L4. We found that the kinase activity of DNA-PKcs is severely inhibited by a commonly used oxygen scavenger system, namely protocatechuic acid (PCA) plus protocatechuate-3,4-dioxygenase (PCD) (Supplementary Fig. 2a, lane 10), which was also used in other smFRET studies for NHEJ synapsis[11,17]. Fortunately, DNA-PKcs can retain its kinase activity when another oxygen scavenger system, specifically glucose plus gloxy (glucose oxidase and catalase) is used (Supplementary Fig. 2b, lane 8). Therefore, the glucose plus gloxy system and 2 mM (±)-6-Hydroxy-2,5,7,8-tetramethylchromane-2-car-boxylic acid (Trolox) (Supplementary Fig. 2b, lane 12) were used in the imaging buffer to stabilize the fluorescence dyes when DNA-PKcs was included in the reaction system. We found similar results for synapsis mediated by Ku and X4L4 when either the PCA plus PCD system, or the glucose plus gloxy system was used in the imaging buffer (Fig. 1 and Supplementary Fig. 3). These include almost all synaptic complexes exhibiting low FRET ($E_{FRET} < 0.6$) (97% vs 98%, Fig. 1d, Supplementary Figs. 1f and 3a, b), equivalent durations of synaptic complexes ($14.3 \pm 0.6$ vs $19.1 \pm 1.8$ s, Fig. 1c and Supplementary Fig. 3c).

DNA-PKcs does not stimulate but slightly inhibits the synapsis efficiency mediated by Ku and X4L4 (Fig. 2a, and Supplementary Fig. 3d), which indicates that DNA-PKcs provides little contribution to the synapsis efficiency. When DNA-PKcs is present in the solution, the $E_{FRET}$ distribution still exhibits a highest peak at ~0.04, and almost no synaptic complex with $E_{FRET}$ larger than 0.6 accumulates (Fig. 2b and Supplementary Fig. 3b). The results are similar to those when DNA-PKcs is absent from the solution (Supplementary Fig. 3a,b). We further tested if DNA-PKcs has an effect on the duration of the synaptic complex formed by Ku and X4L4. The dwell times of synaptic complexes formed with and without DNA-PKcs in the reaction show no significant difference (Fig. 2c, d, and Supplementary Fig. 3c, $p = 0.85$). These results further confirm the negligible role of DNA-PKcs on blunt end synapsis mediated by Ku and X4L4.

When DNA-PKcs is present in the solution, we find that the two dsDNA within the synaptic complex can still move along one another by exhibiting a dynamic trajectory (Supplementary Fig. 3f). This result indicates that the two dsDNA within the synaptic complex continue to be laterally aligned even with DNA-PKcs present. Therefore, the **FS** complex is still formed in reactions containing Ku, X4L4, and DNA-PKcs. To confirm that there was no significant difference between FRET trajectories from synapsis mediated by Ku and X4L4 in the presence and absence of DNA-PKcs, a 1-D autocorrelation function was used to process the dynamic trajectories. The average autocorrelation values were plotted against time to visualize the difference between the dynamics of the trajectories from experiments with and without DNA-PKcs. The fitting of the autocorrelation curves to bi-exponential decays does not show any significant difference from the parameters using an unpaired $t$-test (Supplementary Fig. 3g). These results further confirm that DNA-PKcs has little effect on the **FS** complex.

**XLF drives dsDNA ends into close proximity.** XLF was reported to interact with X4L4 and stimulate X4L4 activity both in vitro and in vivo[10,18]. We therefore tested the effect of XLF on the **FS**. The synapsis efficiency result indicates that XLF significantly increases the efficiency up to 3.6-fold over that without XLF (Supplementary Fig. 4a, lane 2 and lane 1). In addition to the synapsis efficiency stimulation, we checked if XLF causes changes in the synaptic structure by examining $E_{FRET}$ trajectories. As shown in Supplementary Fig. 4c, d, we can observe a pre-dominant population of low $E_{FRET}$ complexes, which has a similar low $E_{FRET}$ level (<0.6) as that observed for synaptic complexes formed by Ku and X4L4 (Fig. 1b, d, e). The dwell time of this low $E_{FRET}$ complex formed with XLF in solution is not significantly different from that formed only by Ku and X4L4 (Supplementary Fig. 4b). This indicates that either XLF is not involved in this low $E_{FRET}$ complex, or that XLF has little effect on this synaptic complex. In the same XLF experiment, a smaller population of synaptic complexes exhibit an additional $E_{FRET}$ state with a value ~0.8 as shown on the representative traces (Fig. 3a, b and Supplementary Fig. 4e). The appearance of a high $E_{FRET}$ synaptic complex indicates that XLF indeed has a large effect on a sub-population of synaptic complexes formed by Ku and X4L4. Consistent with the existence of two different major $E_{FRET}$ synaptic complexes, the total $E_{FRET}$ distribution of the synaptic complexes by Ku, X4L4, and XLF shows two clear main peaks (Fig. 3d). The higher $E_{FRET}$ peak stimulated by XLF has a value of ~0.8 (Fig. 3d), which is the same as the $E_{FRET}$ obtained from a pre-ligated positive control (Supplementary Fig. 4f). Unlike the **FS** complex formed with Ku and X4L4, within which the two duplexes are only brought into lateral configuration (Fig. 1a), the two dsDNA within this XLF-stimulated high FRET synaptic complex can contact each other in an end-to-end configuration (Fig. 3d). This transition is also confirmed by a direct observation of transition from an initial low $E_{FRET}$ to a stable high $E_{FRET}$ state (~0.8) (Fig. 3a and Supplementary Fig. 4e). Therefore, we refer to this stable synaptic state with an $E_{FRET}$ peak of ~0.8 as close synapsis (**CS**). The dwell time of XLF-stimulated **CS** complex is $21.2 (\pm 4.6)$ s (Fig. 3e).

The **CS** complex can be formed in two different ways (Fig. 3a, b). In the first one, there is a transition from low $E_{FRET}$ **FS** complex to high $E_{FRET}$ **CS** complex after some delay (Fig. 3a and Supplementary Fig. 4e). This suggests that XLF binds to the **FS** complex that was formed by Ku and X4L4 and then aligns the two ends within the **FS** complex to form the **CS** complex. The lag time of this transition is $0.93 (\pm 0.06)$ s (Fig. 3c). For the second case, the high $E_{FRET}$ **CS** complex is immediately formed (Fig. 3b),

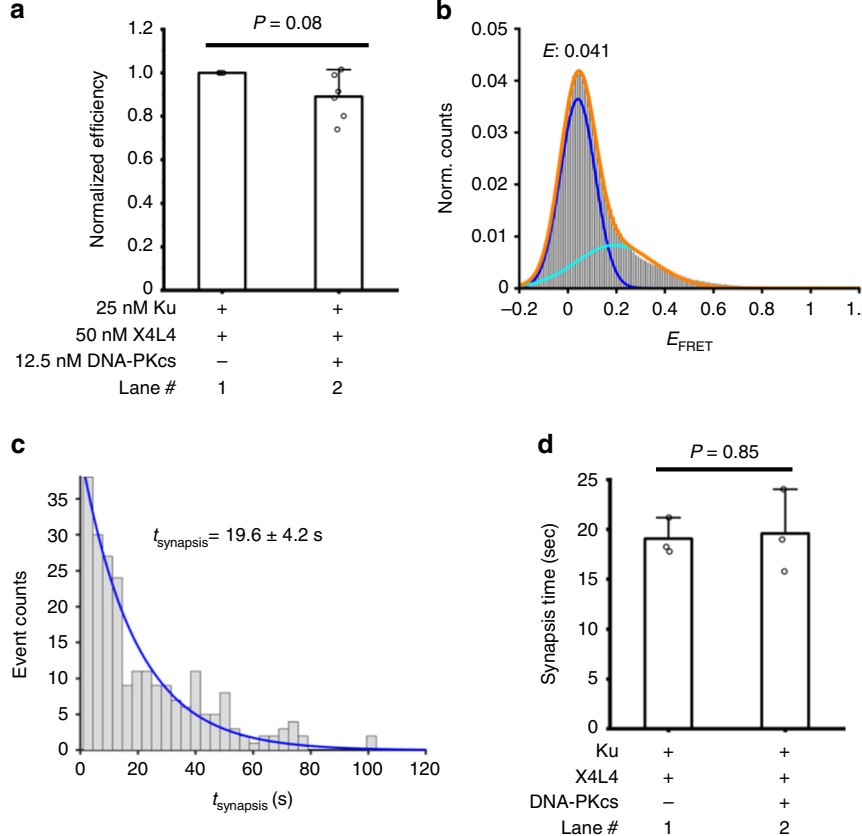

**Fig. 2** DNA-PKcs has little effect on synapsis mediated by Ku plus X4L4. **a** Normalized synapsis efficiency mediated by different NHEJ factors −25 nM Ku, 50 nM X4L4, and 12.5 nM DNA-PKcs using the glucose plus gloxy oxygen scavenger system. Data are represented as mean ± SD of at least four independent replicates. $T$-test (unpaired, two-tailed) was applied for $p$ value calculation ($p = 0.08$). **b** Histogram of $E_{FRET}$ values of all synapsis events mediated by 25 nM Ku, 50 nM X4L4, and 12.5 nM DNA-PKcs using the glucose plus gloxy oxygen scavenger system in the solution. The $E$ value shown on the distribution was obtained by a Gaussian fit of the highest peak. $n = 792$ molecules. **c** Histogram and corresponding exponential fit of total synapsis time mediated by Ku, X4L4, and DNA-PKcs using the glucose plus gloxy oxygen scavenger system in the solution. Cy3 signal lifetime (including zero-FRET and detectable FRET portions) of the synaptic complex was used to calculate the total dwell time for each synapsis event, and only the synapsis events ($n = 231$) with both start and end time points within the detection time window were included. Synapsis time shown on graph is represented as mean ± SD of three replicates. **d** Summarized dwell times of synaptic complexes formed by Ku and X4L4, and by Ku, X4L4, and DNA-PKcs using glucose plus gloxy oxygen scavenger system in the solution. Error bars represent SD of three replicates. $T$-test (unpaired, two-tailed) was applied for $p$ value calculation. The corresponding dwell distributions and exponential fits are shown in **c** and Supplementary Fig. 3c. Source data are provided as a Source Data1 file

which suggests that the **CS** complex is formed by XLF, Ku, and X4L4 together with no delay within our detection time resolution. These results indicate that XLF can drive the two dsDNA ends into close synapsis via either a stepwise or a nearly single-step assembly of these proteins.

We also tested the possibility of lateral alignment of two duplexes within the synaptic complex when XLF is present in the reaction. The number of detected nonzero FRET molecules for the midpoint-labeled configuration is only 26% of that for the end-labeled configuration (Supplementary Fig. 4g). This result indicates that the two dsDNA molecules interact predominately at the DNA ends when XLF is present in the reaction.

We were interested in testing whether formation of the **CS** complex would show a strong or weak dependence on XLF protein concentration. We find that with an 18-fold increase in XLF concentration (from 16.7 nM to 300 nM), there is only a threefold increase (from 17.4% at 16.7 nM XLF to 48.4 % at 300 nM XLF) in **CS** formation (Fig. 3f and Supplementary Fig. 5). It has been reported that XLF mediates synapsis of two dsDNA molecules either via filament formation or via direct end-to-end interaction[8,11,13,17,19–22]. We would anticipate that any XLF filament formation would show an exponential dependence on XLF concentration, and this was not the case. This suggests

that XLF is primarily acting at the DNA ends rather than by forming a multi-protein filament along the length of the DNA. Moreover, XLF promotes the formation of the **CS** complex, where the two DNA molecules are in end-to-end contact, which further argues that the major contribution of XLF is at the DNA DSB ends.

**PAXX can modestly promote CS complex formation.** PAXX is the most recently identified NHEJ factor. PAXX, which forms a homodimer in solution, has structural similarity to X4 and XLF, and was reported to promote Ku-dependent DNA ligation[23–25]. We also examined the effect of PAXX on synapsis. We find that though PAXX does not impact the total synapsis efficiency mediated by Ku and X4L4 (Supplementary Fig. 4a, lane 1 and lane 3), it can indeed stimulate the formation of a high $E_{FRET}$ synaptic complex as shown on the representative traces (Fig. 4a and Supplementary Fig. 6a,b). The total $E_{FRET}$ distribution also exhibits a high $E_{FRET}$ peak in addition to the low $E_{FRET}$ (<0.6) **FS** complex (Fig. 4b). When fit to a Gaussian function, the high $E_{FRET}$ synaptic complex exhibits an $E_{FRET}$ of ~0.77 (Fig. 4c), which is similar to the $E_{FRET}$ level of the pre-ligated positive control. These results indicate that PAXX can also drive dsDNA

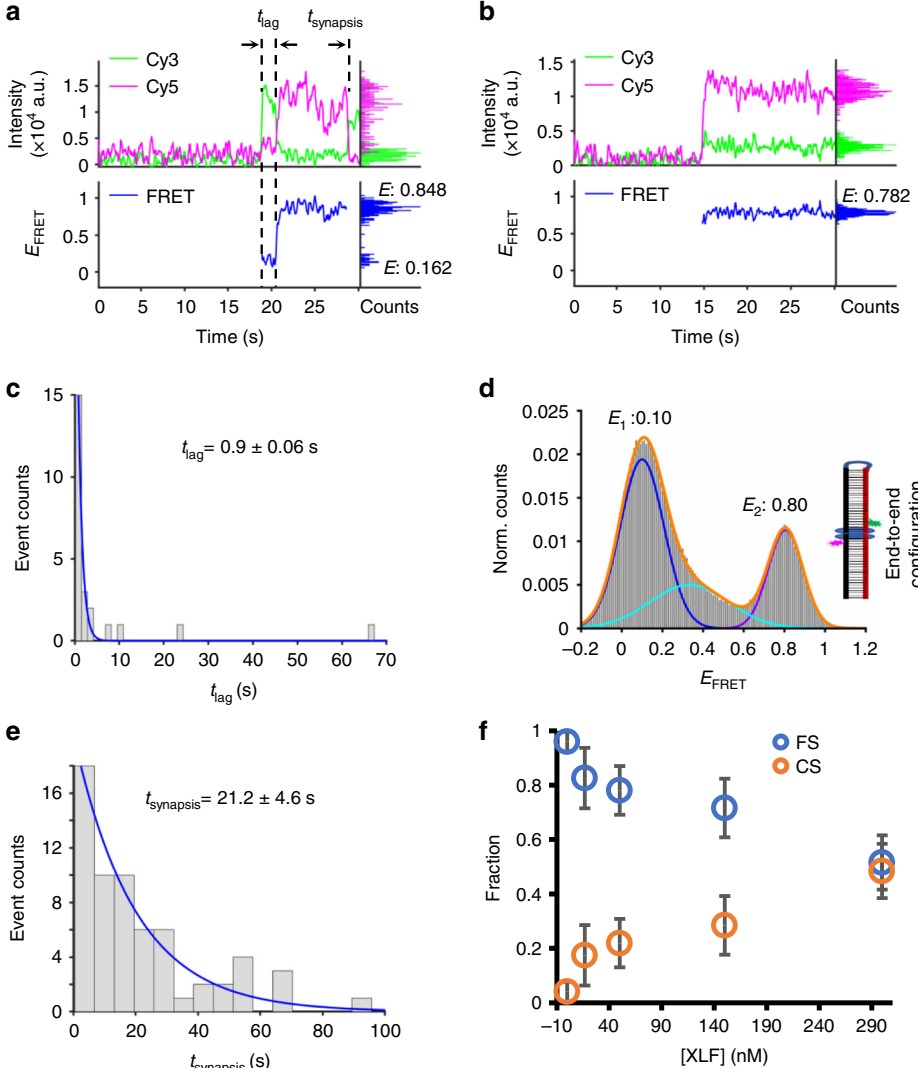

**Fig. 3** XLF increases end proximity of dsDNA within synaptic complexes. **a**, **b** Representative single-molecule time traces of donor (green) intensity, acceptor (magenta) intensity, and corresponding $E_{FRET}$ values (blue) for synapsis mediated by 25 nM Ku, 50 nM X4L4, and 50 nM XLF. The right parts are histograms of donor intensity (green), acceptor intensity (magenta), and $E_{FRET}$ values (blue) within synapsis period. **c** Histogram and corresponding exponential fit of lag time between synapsis starting and transition to high $E_{FRET}$. Only the events with a detectable transition from low $E_{FRET}$ ($E < 0.6$) to high $E_{FRET}$ ($E \geq 0.6$) were included. $n = 23$ traces. Error represents the SEM of the fit. **d** Histogram of $E_{FRET}$ values of all synapsis events mediated by Ku, X4L4, and XLF. The $E$ value shown was obtained from a Gaussian fit of the highest peak of each kind of synaptic complex (**FS** or **CS**). $n = 423$ molecules. The diagram illustrates an end-to-end configuration of the **CS** state which is ready for ligation. **e** Histogram and corresponding exponential fit of synapsis time of high $E_{FRET}$ ($E \geq 0.6$) events mediated by Ku, X4L4, and XLF. Only the high $E_{FRET}$ events ($n = 63$) with both start and end time points within the detection time window were included. Error represents the SD of two independent replicates. **f** XLF concentration-dependent synaptic complex formation. The reaction contains 25 nM Ku, 50 nM X4L4, and varied XLF. **FS** complex: $E_{FRET} < 0.6$, **CS** complex: $E_{FRET} \geq 0.6$. Data are represented as mean ± SD of three independent replicates. Source data are provided as a Source Data1 file

ends into close proximity to form the **CS** complex. Similar to XLF, PAXX does not have a significant effect on the duration of the **FS** complex (Supplementary Fig. 4b). Furthermore, like XLF, PAXX can bind stepwise to the **FS** complex (Supplementary Fig. 6b) to form the **CS** complex or form the **CS** complex together with Ku and X4L4 simultaneously (Fig. 4a and Supplementary Fig. 6a). Under these conditions, the ratio of the **CS** complex stimulated by PAXX (15.3%) is lower than that stimulated by XLF (34.1%) (Fig. 4f, lane 1 and lane 2), and the duration of **CS** complex promoted by PAXX is 5.5 (±0.07) s (Fig. 4d and Supplementary Fig. 6e), which is significantly shorter than that by XLF (21.2 (±4.6) s) (Fig. 3e). These results demonstrate that PAXX can stimulate **CS**, but with a relatively weak strength and modest synapsis efficiency.

**XLF and PAXX independently stimulate CS complex formation.** A previous study showed that PAXX and XLF have overlapping functions in ligation[10]. We tested whether XLF and PAXX function synergistically or independently at the synapsis step, by observing the DNA synapsis with XLF and PAXX both added to the Ku plus X4L4 system. The total FRET of the synaptic complex stimulated by XLF and PAAX together in the solution still exhibits two main peaks (Fig. 4e). One has a low $E_{FRET}$ of 0.08, and the other has a high $E_{FRET}$ of 0.82. The $E_{FRET}$ levels of the two major peaks are similar to those seen when XLF or PAXX is added individually (Figs. 3d, 4b). These results indicate that XLF and PAXX do not affect one another for synaptic complex formation. They also do not affect each other for **CS** complex formation, whether this is via stepwise formation (namely,

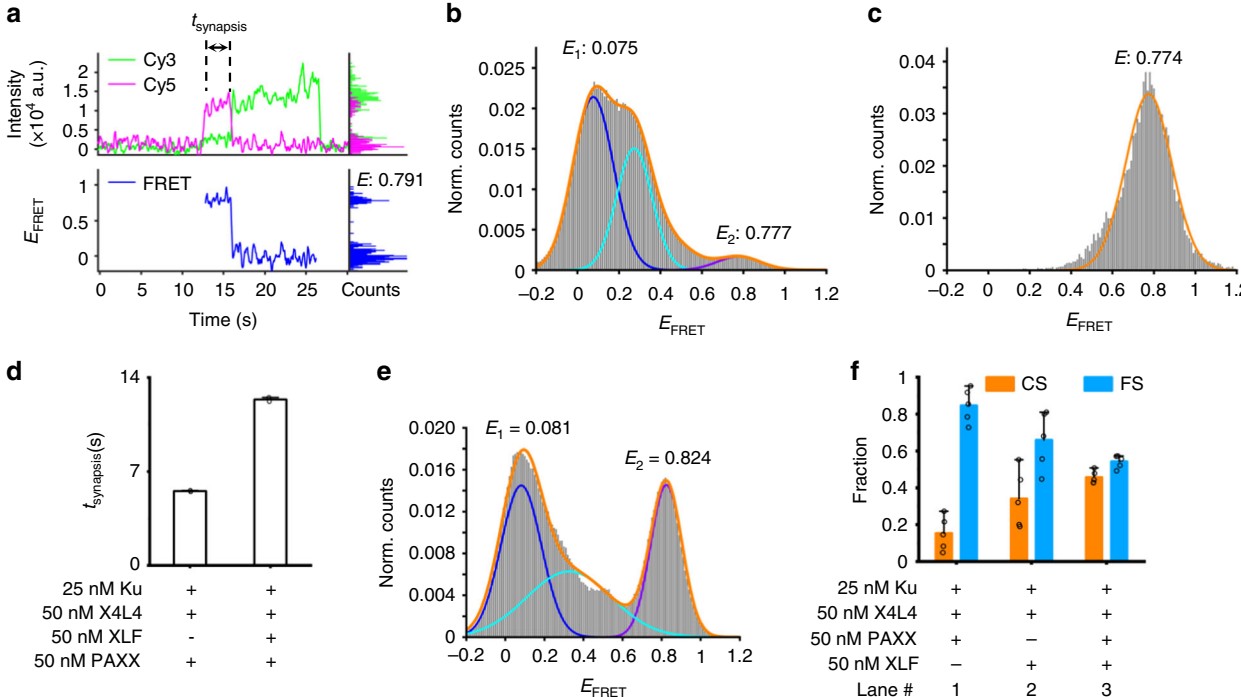

**Fig. 4** PAXX increases end proximity of two dsDNA but with modest efficiency. **a** Representative single-molecule time traces of donor (green) intensity, acceptor (magenta) intensity, and corresponding $E_{FRET}$ values (blue) for synapsis mediated by 25 nM Ku, 50 nM X4L4, and 50 nM PAXX. The right parts are histograms of donor intensity (green), acceptor intensity (magenta), and $E$ values (blue) within the synapsis period. **b** Histogram of $E_{FRET}$ values of all synapsis events mediated by Ku, X4L4, and PAXX. The $E$ value shown was obtained from a Gaussian fit of the highest peak of each kind of synaptic complex (**FS** and **CS**). $n = 223$ molecules. **c** Histogram of $E_{FRET}$ values of all synapsis events with $E_{FRET} \geq 0.6$. The $E$ value was obtained from a Gaussian fit of the corresponding peak. $n = 33$ events. The data used here is the same batch as that used in **b**. **d** Dwell time of **CS** complex stimulated by PAXX alone or XLF plus PAXX. Only the high $E_{FRET}$ ($E \geq 0.6$) events with both start and end time points within the detection time window were included. Error bars represent SD of two replicates. The corresponding dwell distributions and exponential fits are shown in Supplementary Figs. 6e, f. **e** Histogram of $E_{FRET}$ values of all synapsis events mediated by Ku, X4L4, XLF, and PAXX. The $E$ value shown was obtained from a Gaussian fit of the highest peak of each type of synaptic complex (**FS** and **CS**). $n = 429$ molecules. **f** Fraction of low $E_{FRET}$ (**FS**) complex and high $E_{FRET}$ (**CS**) complex mediated by different combinations of NHEJ factors. Data are represented as the mean ± SD of at least three replicates. Source data are provided as a Source Data1 file

transition from low $E_{FRET}$ to high $E_{FRET}$) or a single-step formation of the high $E_{FRET}$ complex (Supplementary Fig. 6c,d). These results indicate that XLF and PAXX function through similar interactions with Ku and/or X4L4 proteins to promote **CS** complex formation, consistent with the fact that they are paralogs.

When XLF and PAXX are present together in the same reaction, there is a slight increase in the fraction in the **CS** state (Fig. 4f, lane 2 to lane 3). This indicates that XLF and PAXX both contribute to synapsis in this mixed reaction system. These two factors could conceivably increase the **CS** complex independently within different synaptic complexes or synergistically within the same complex. If they function synergistically, we would expect a more stable **CS** complex which lasts longer, leading to accumulation of the **CS** complex. Moreover, we would expect a higher transition rate from low $E_{FRET}$ **FS** to high $E_{FRET}$ **CS** for the synergistic case, if the **CS** forms via a stepwise assembly. The lag time of the transition from **FS** to **CS** in the presence of both XLF and PAXX is 2.35 (±0.17) s (Supplementary Fig. 6g). This is longer than the lag time observed in the presence of only XLF (0.93 (±0.06) s) (Fig. 3c). This indicates that PAXX impairs the transition rate in the system when both XLF and PAXX are present. The dwell time (12.3 (±0.2) s) of the mixed **CS** complex formed with both XLF and PAXX present (Fig. 4d and Supplementary Fig. 6f) is shorter than the time (21.2 (±4.6) s) due to XLF stimulation alone (Fig. 3e). These results indicate that XLF and PAXX independently stimulate **CS** complex formation. Both proteins contribute to the fraction of **CS** complex formed,

while the shorter lifetime of the **CS** complex formed by PAXX compromises the lifetime of the **CS** complexes.

We also tested the effect of DNA-PKcs on the synapsis stimulated by PAXX or XLF. DNA-PKcs has little effect on the total synapsis efficiency (**FS** + **CS**) mediated by Ku, X4L4, and PAXX (Supplementary Fig. 7a), while DNA-PKcs reduces the total synapsis efficiency mediated by Ku, X4L4, and XLF (Supplementary Fig. 7b). When DNA-PKcs is present in the solution, it reduces the formation of **CS** complex generated by Ku, X4L4, and PAXX, or by Ku, X4L4, and XLF (Supplementary Figs. 7c-g). This small inhibition could be because the binding of DNA-PKcs at the dsDNA ends without quick dissociation within some synaptic complexes may push the Ku protein internally along the DNA duplex. This might conceivably block the accessibility of Ku for XLF and PAXX.

**The CS complex accumulates with time.** Previous studies have shown that DSBs are generally repaired in vivo with a half-life of 10–60 min[26–28], and most of the DSBs are repaired via the NHEJ pathway[26]. We examined the time-dependent formation of the **CS** complex (which is ready for dsDNA end ligation) to determine if the synapsis kinetics observed here are relevant to the physiological NHEJ repair kinetics. The total synapsis efficiency including **FS** and **CS** complexes increases with reaction time and reaches a maximum at ~15 min for both reactions, namely with XLF present (Supplementary Fig. 8a) or with XLF and PAXX both present (Supplementary Fig. 8b) in the system. The **CS**

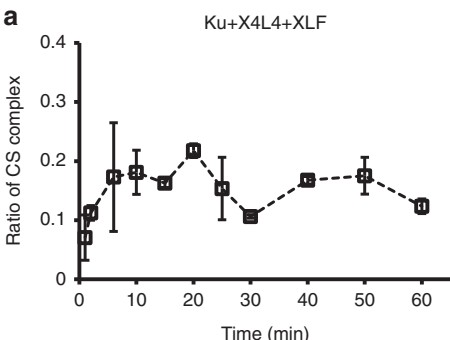
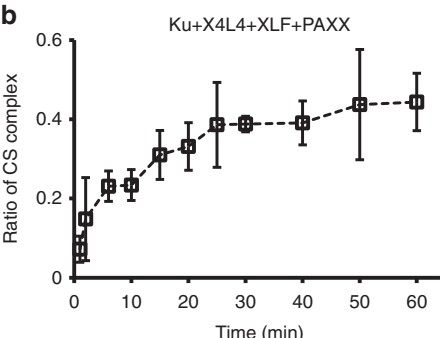

**Fig. 5** Time-dependent accumulation of CS complex. **a**, **b** Kinetics of **CS** complex formation with 25 nM Ku, 50 nM X4L4, and 50 nM XLF (**a**) or with 25 nM Ku, 50 nM X4L4, 50 nM XLF, and 50 nM PAXX (**b**). Data are represented as the mean ± SD of two independent replicates. Source data are provided as a Source Data1 file

complex fraction also increases with time and reaches a plateau at ~20 min when Ku, X4L4, and XLF are present (Fig. 5a), and 25 min when Ku, X4L4, XLF, and PAXX are present (Fig. 5b). Therefore, the kinetics of **CS** complex formation observed here is similar to that seen within living cells.

## Discussion

Bringing together two broken DNA ends in the necessary translational and rotational orientation and with sufficient spatial flexibility to permit nuclease and polymerase modification of each end for repair is one of the most demanding steps in molecular biology. This noncovalent step is the major kinetic and physical barrier, and without synapsis, a major chromosomal structural alteration will result (deletion, translocation, inversion, or duplication). Our approach here has been to fully purify several key human NHEJ proteins and study them using smFRET.

Our results identify two major structural complexes formed during NHEJ synapsis (Fig. 6b). Ku and X4L4 mediate synapsis to form a flexible synaptic (**FS**) complex. In the **FS** complex, the two dsDNA are merely brought into a lateral (parallel side-by-side) duplex proximity that is close to but not necessarily directly at the DNA ends. The data suggest that the two dsDNA duplexes within the **FS** complex can still move along each other (in a side-by-side manner). In the **FS** complex, X4L4 can bind to each dsDNA through the interaction of L4 BRCT domains with the Ku heterodimer[29] and/or the interaction of L4 with the dsDNA[13,30]. Then the two dsDNAs can be brought together through the interaction of X4[21], which bridges the two Ku-X4L4-DNA complexes, bringing the two dsDNAs into a lateral duplex configuration (Fig. 6b). Given the potential for X4 to form oligomers both in solution and upon binding to dsDNA[21,22], as well as the strong binding of L4 to the flexible 130 Å tail of X4[1,19,31], there are many noncovalent intermediate states that can be envisioned during the interaction of the two dsDNA duplexes (Fig. 6b).

Our results show that docking the two dsDNAs within the **FS** complex into an end-to-end configuration requires either the XLF or PAXX protein, with XLF playing a more important role (Fig. 6b). XLF can interact with X4 through a head-to-head interaction[18,19,32]. Also, it was reported that XLF and PAXX can interact with Ku[10,33]. These interactions can position the two DNA ends closer together, thus forming a high $E_{FRET}$ (**CS**) synaptic complex (Fig. 6b).

According to our synapsis model proposed above, we would predict that it is difficult for the two dsDNA ends within the **FS** complex to be covalently ligated because of the suboptimal orientation, while the ends within the **CS** complex are readily ligated because they are aligned in an appropriate end-to-end

orientation and are in close contact. Importantly, XLF (Fig. 6a and Supplementary Fig. 9, lane 10) and/or PAXX (Fig. 6a and Supplementary Fig. 9, lane 11 and lane 12), which promote the formation of **CS** complex, can indeed stimulate the blunt end ligation mediated by Ku plus X4L4 in our ensemble ligation assay. In contrast, Ku plus X4L4 (Fig. 6a and Supplementary Fig. 9, lane 9) or Ku, DNA-PKcs, and X4L4 (Fig. 6a and Supplementary Fig. 9, lane 7), which can only promote **FS**, cannot mediate detectable ligation of blunt ends.

XLF stimulates X4L4 activity for DNA ends joining, but the mechanistic basis for this stimulation has not been clear[18,34]. Our results here show that XLF can increase the synapsis efficiency. Moreover, XLF can stabilize the synaptic complexes and increase the proximity of the two blunt DNA ends. These results can explain quite well the mechanism of XLF stimulation of L4 activity. A recently published study shows that disrupting the domains of XLF either for X4 interaction or for Ku binding impairs NHEJ blunt end repair of the type that lacks nucleotide insertion or deletion (indels) in cells; and double mutants disrupted for both of these binding domains show a nearly complete loss of this type of blunt end NHEJ[35]. This supports our finding that XLF, by interacting with X4 and/or Ku, aids the synapsis mediated by Ku and X4L4. The tethering of DNA ends together described in our study here might protect the ends from further processing by nucleases and polymerases. This may explain why XLF facilitates the end joining of blunt ends[35]. Once the high $E_{FRET}$ synaptic complex (**CS**) forms, it is relatively stable compared with the low $E_{FRET}$ **FS** complexes. This likely facilitates end protection and ligation.

Recent work suggests that a single XLF dimer in the synaptic complex bridges the two dsDNA ends[17], and the interaction of XLF with X4 is required for the bridging. This is contrary to the in vivo results just mentioned above showing that only double mutants in XLF having both the binding domains for X4 and Ku disrupted show a complete loss of NHEJ without indels[35] and that the X4–XLF interaction is variably required for DNA repair[36]. Furthermore, our results here show that addition of stoichiometric or greater amounts of XLF promotes somewhat more **CS** complex formation, suggesting that one or a few XLF dimers may be enough to stimulate the **CS** complex. We note that PAXX can also drive the two dsDNA ends together, and PAXX is not known to interact with X4[2] (Supplementary Discussion). Moreover, the formation of **CS** complex stimulated by XLF suggests that the synapsis is primarily end-to-end (Supplementary Discussion). Combining our findings here with known interactions, we propose that up to three XLF dimers at the DNA ends may simultaneously stabilize the **CS** complex to allow more **CS** complex accumulation (Fig. 6b).

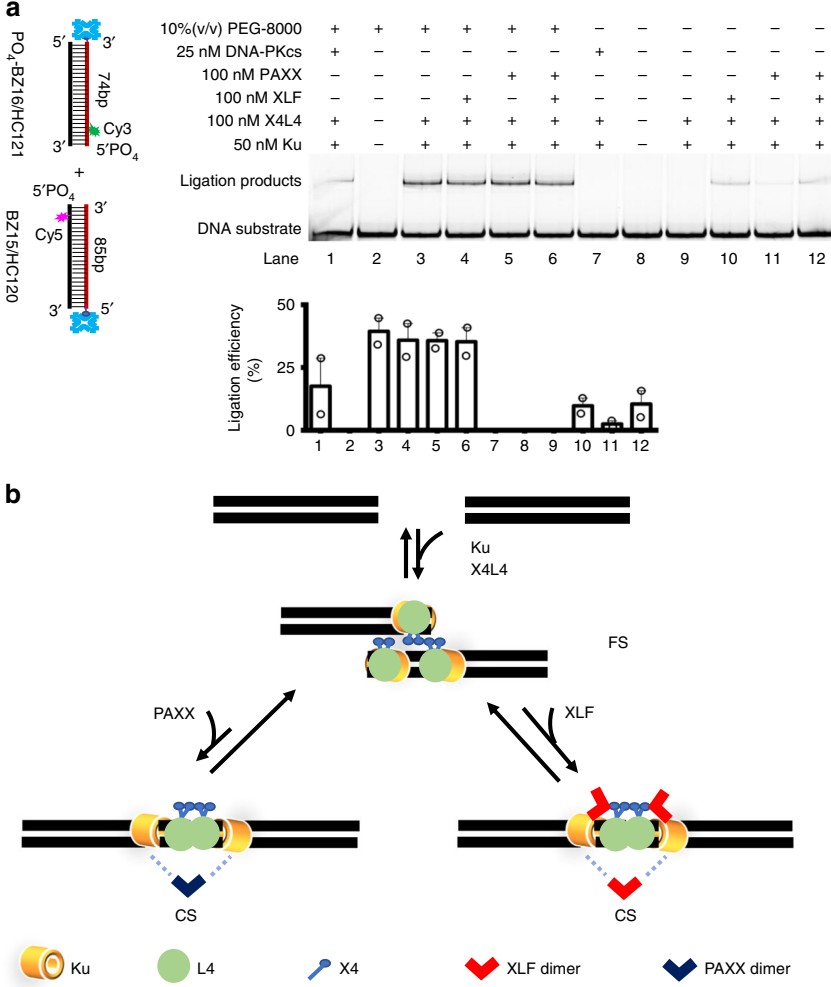

**Fig. 6** Ensemble confirmation and the proposed model of NHEJ synapsis. **a** (Top) Gel results of blunt end ligation mediated by different combinations of NHEJ components in the presence and absence of 10% (v/v) PEG-8000. (Bottom) Quantified ligation efficiency. Reactions 1–12 correspond to the reactions represented by lanes 1–12 in the top panel, respectively. Data are represented as mean ± SD of two independent replicates. PEG-8000 is a volume excluder, which increases the collision frequency of the two dsDNA together ready for covalent ligation. With this volume excluder in the solution, the ligation efficiency was similar with or without stimulation from XLF, PAXX, or both (lanes 3–6). Without the volume excluder in the solution, where only NHEJ proteins would be available to bridge the two dsDNA ends for ligation, we find that Ku and X4L4 cannot mediate the ligation of blunt end dsDNA (lane 9), and XLF or PAXX can stimulate the covalent ligation (lanes 10–12). BZ15 was synthesized with a 5' PO$_4$. **b** Ku and X4L4 mediate a flexible synapsis (**FS**), in which two dsDNA are brought into a lateral configuration. Aligning the two dsDNA within the **FS** complex to an end-to-end state is stimulated by the XLF or PAXX protein, and XLF is the more efficient one in this respect (as reflected by the length of the reaction arrow). One XLF dimer either interacting with X4 in X4L4 or with Ku may be sufficient for close synapsis (**CS**), but we speculate that more than one XLF dimer binding to both Ku and X4 may more fully stabilize the **CS** complex. We have not shown filament formation in this diagram, but for chromatinized DNA templates, it is possible that filaments may be important[13]. Source data are provided as Source Data1 and Source Data2 files

In conclusion our results show that Ku plus X4L4 are necessary and sufficient for a flexible synapsis (**FS**) of blunt end dsDNA, and that XLF can promote a transition from **FS** to close synapsis (**CS**), whereas DNA-PKcs is not required for synapsis (Supplementary Discussion). The definition of **FS** and **CS** states using the simplest purified NHEJ components provides a sound basis for the NHEJ field to move forward with more complex cases, such as overhanging ends that are compatible or incompatible. The blunt ends examined here are the most challenging case for synapsis because there is no base pairing between the DNA ends. We do not preclude that additional conformations will be elucidated with future progress, particularly as this knowledge is integrated with cryo-EM and atomic structural insights from many methodologies applied to this highly important molecular process. But it is of key importance that we have established the essential elements for synapsis of two DNA ends during NHEJ.

## Methods

**DNA probes.** All ssDNA oligonucleotides (oligos) were synthesized by IDT and sequences of these oligos are summarized in Supplementary Table 1. The fluorophore Cy3/Cy5-labeled oligos were purified by IDT using the HPLC method, and directly used without further purification. All other oligos were purified in the lab using urea-denaturing PAGE. Because BZ35 and BZ15 were synthesized with a phosphate group at the 5' end, they were treated with shrimp alkaline phosphatase (rSAP, NEB, and M0371S) at 37 °C overnight to remove the phosphate group. The 5'OH-BZ35 and 5'OH-BZ15 were finally recovered by phenol extraction followed by ethanol precipitation. The duplexes used for immobilization were obtained by annealing of Cy5-labeled oligo (5'OH-BZ15 or BZ69) and HC120 in buffer of 20 mM Tris-HCl, 100 mM NaCl, pH 8.0. Oligo HC120 has a biotin at the 5' end for immobilization on slide surface. The secondary 'looped' Cy3-labeled duplex (Cy3-loop) was obtained by ligation of a stem-loop structured BZ24 with an annealed short duplex 5'OH-BZ35/BZ52 using T4 DNA ligase (Millipore Sigma, 10716359001) at 15 °C overnight. The target ligation product, called looped duplex, was then isolated from input short duplexes and self-ligated products using 12% native PAGE. The pre-ligated positive control was obtained by ligating BZ15/HC120 and phosphorylated Cy3-looped duplex using Ku, X4L4, and XLF at 37 °C for 1 h.

**Protein expression and purification**. Ku70/80 heterodimer, DNA-PKcs, and PAXX were the same purified batches as used in a previous study and documented to be functional[9]. L4 with an N-terminal snap-tag and a C-terminal 9Xhis-tag was co-expressed with X4 in High Five insect cells (Invitrogen, B855-02). After thawing on ice, the cell pellets infected with X4 and L4 baculovirus were lysed with buffer of 50 mM sodium phosphate, 500 mM KCl, 0.1% Triton X-100, 20 mM imidazole, 2 mM β-ME, 2 μg/ml aprotinin, 1 μg/ml pepstatin A, 10 μM leupeptin, 1 μM bestatin, and 0.2 mM PMSF, pH 7.8. Then the cell lysate was sequentially passed through Ni-NTA affinity chromatography, anion exchange chromatography and cation exchange chromatography. After cation exchange Mono S column, fractions containing X4L4 were pooled, aliquoted, frozen down using liquid nitrogen, and stored at −80 °C. XLF was expressed and purified using the method reported previously[37]. Briefly, XLF cDNA on a pcDNA6 plasmid (pcDNA6/huXLF-Myc-His) was transiently transfected into 293 T cells using the PEI method. The cells were then harvested 48 h after transfection. Ni-NTA chromatography and anion exchange chromatography were then used for XLF purification. The protein concentration was measured by SDS-PAGE gel using BSA as standard.

**DNA-PKcs kinase activity test**. Briefly, 20 nM DNA-PKcs and 80 nM [γ-$^{32}$P] ATP with or without 1 μM GW21/22 (22 bp) dsDNA were incubated in different buffer conditions as specified on the graphs. Buffer 1, which consists of 25 mM Tris-HCl, pH 7.5, 10 mM MgCl$_2$, 10 mM DTT, and 5% sucrose, was used as a positive control. The reaction was conducted at 37 °C for 10 min. Then the reaction was stopped by adding 6 × SDS loading dye and boiled at 100 °C for 5 min. The radioactive products were analyzed by 15% SDS-PAGE and autoradiography.

**Ensemble ligation reaction**. The activity of purified X4:snap-L4 was tested as previously described[9] with minor modification. Briefly, 40 nM BZ15/HC116, 40 nM 5′-PO$_4$-BZ16/BZ18, 0.5 mM ATP, 1 mM DTT, 10% PEG-8000, 50 nM Ku70/80, and 100 nM X4L4/X4:snap-L4 were sequentially added into the ligation buffer consisting of 25 mM Tris-HCl, pH 8.0, 75 mM KCl, and 10 mM MgCl$_2$. The reaction solution was mixed well and incubated at 37 °C for 1 h. The mixture then went through phenol extraction and ethanol precipitation. After that, the ligation products were resolved using a 12% native PAGE, which was then imaged using a Typhoon FLA9500 instrument using the laser specific for Cy5 excitation.

The ligation reactions stimulated by different NHEJ components as shown on Fig. 6a and Supplementary Fig. 9 were conducted in reaction buffer of 20 mM Trisacetate, pH 7.5, 75 mM KAc, and 10 mM MgCl$_2$. In all, 40 nM BZ15/HC120 and 40 nM 5′-PO$_4$-BZ16/HC121 were first incubated with 0.1 mg/ml neutravidin in reaction buffer for 5 min. Then 0.5 mM ATP, 1 mM DTT, 10% PEG-8000, 50 nM Ku70/80, 25 nM DNA-PKcs, 100 nM XLF, 100 nM PAXX, and 100 nM X4:snap-L4 were sequentially added into the DNA mixture as specified on the figures. The reaction solution was mixed well and incubated at 37 °C for 1 h. The remaining steps were the same as that described above for X4L4 activity test.

**Slide and coverslip modification**. Slides and coverslips for imaging were modified as previously described[38,39] with minor modifications. Briefly, slides and coverslips were first treated with 4 M NaOH using sonication for 30 min After thoroughly washing with distilled water, the slides and coverslips were subjected to a 'piranha' solution (three volumes of H$_2$SO$_4$ (98% v/v) to one volume of H$_2$O$_2$ (30% v/v)) for another 30 min, followed by distilled water washing ten times and methanol washing twice. The slides and coverslips then stayed in HPLC-grade methanol until silanization, in which methanol solution containing 2% (v/v) (3-aminopropyl-triethoxy)silane (Millipore Sigma, 440140) and 5% (v/v) acetic acid was used to functionalize the slide and coverslip with primary amine groups. After washing with water then drying, slides and coverslips were PEGylated by using a viscous solution of mPEG (Laysan Bio, mPEG-SVA-5000) and a mixture of mPEG and Biotin-PEG (Laysan Bio, Biotin-PEG-SVA-5000), respectively. The ratio of mPEG to Bitotin-PEG was set to 40:1 for coverslip. PEGylation was done at room temperature in the dark overnight. The slides and coverslips were then thoroughly washed with distilled water, dried, and kept at −20 °C until use.

**Microscopy setup and single-molecule image acquisition**. Single-molecule experiments were done on a total internal reflection fluorescence microscope equipped on an inverted Nikon Eclipse Ti microscope with a ×100, 1.49 NA oil immersion objective lens (Cell and Tissue Imaging Core, USC). The 514 nm and 647 nm lasers were coupled into AOTF and introduced into the microscope via a fiber. A dichroic mirror (ZT532/640rpc-UF3, Chroma) in the microscope turret, was used to reflect lasers into the objective and let emission fluorescence pass through. The emission fluorescence was then split into two channels using a second dichroic mirror (FF640-FDi01-25x36, Semrock) mounted into a TuCam (Andor). The split donor and acceptor emission then passed through a narrowband band-pass filter (Chroma ET600/50 m for green channel; Chroma ET700/75 m for red channel) and collected by two perpendicularly mounted Ixon + 897 Ultra EMCCD cameras (Andor). Each day before experiments, we always imaged the diffraction-limited fluorescent beads (Invitrogen) that have wide emission spectra spanning both channels to accurately register the two CCD cameras. The location of the beads was also used to map the two channels for data analysis.

**Sample preparation for single-molecule imaging**. Unless otherwise specified, all reactions were carried out at room temperature (22 °C) with a buffer containing 20 mM Tris-acetate pH 7.5, 75 mM KCl, 10 mM MgCl$_2$, 3% glycerol, 1 mg/ml BSA, 2 mM DTT, 0.5 mM ATP, 0.25% Tween 20, 5 mM Trolox (Millipore Sigma, 238813), and an oxygen scavenger system [5 mM PCA (Millipore Sigma, 03930590) and 0.42 U/ml PCD, OYC Americas Inc. 46852004][40,41]. Since the PCA plus PCD oxygen scavenger system severely reduces the kinase activity of DNA-PKcs, all reactions involving DNA-PKcs were carried out using the above buffer condition but with reduced Trolox (2 mM), and with another oxygen scavenger system [0.8% (w/v) D-glucose (RPI Research Products International, G32040-5000) and 1× gloxy consisting of 165 U/ml glucose oxidase (Millipore Sigma, G2133-50KU), and 2170 U/ml catalase (Millipore Sigma, C3556)]. For all reactions, 250 pM 5′OH-(Cy5)-BZ15/HC120 or BZ69/HC120 dsDNA as specified on the figures was immobilized on the coverslip surface via the biotin-neutravidin-biotin strategy, and 10 nM Cy3-labeled loop-structure 5′OH-dsDNA was used in the solution. The final protein concentration applied in the study was as follows unless otherwise specified: 25 nM Ku, 50 nM X4L4, 12.5 nM DNA-PKcs, 50 nM XLF, and 50 nM PAXX. Proteins and looped dsDNA were added stepwise into the reaction buffer and quickly mixed. The reaction mixture was then immediately injected into the imaging chamber which was made with a slide and a coverslip with Cy5-DNA already immobilized. Images were captured immediately after sample injection or after incubation for certain times as denoted in the figure legends. CCD camera exposure time was set at 50 ms. At least three movies were captured for each sample and each movie lasted about 2 min, unless otherwise specified. At least two independent samples were analyzed for each condition.

**Single-molecule imaging for reactions with different times**. To monitor the kinetics of **CS** complex formation as shown in Fig. 5, the same sample was monitored after each reaction for a specific amount of time. To avoid the bleaching of fluorescence dyes resulting from prolonged illumination, each region of interest (ROI) was only imaged for 20 s, then a new ROI region was quickly switched manually and imaged for another 20 s. Three movies with 20-s durations were captured for each time point. All molecules identified in these three movies were combined to calculate the fraction of the **CS** complexes in all the identified complexes for a specific time point. The normalized synapsis efficiency shown in Supplementary Fig. 8 was calculated by normalizing the molecules obtained at a specific time to all the molecules collected from all the time points.

**smFRET data analysis**. All FRET pairs and corresponding single-molecule trajectories were searched and extracted using iSMS v2.01 software[42,43] built in MATLAB. The FRET pairs were searched by scanning every ten frames of the movie. The final FRET pair number was obtained after visual identification of all the trajectories. The molecules whose signals last the whole movie were not selected unless at least one anticorrelated transition between Cy3 and Cy5 signals was observed. Since more than 90% of the identified molecules only exhibited one synapsis event within the capturing time, the total synapsis efficiency was calculated from the number of final FRET pairs identified (subtracted by the nonspecific FRET pairs from the reaction only with dsDNA and without NHEJ proteins present). Because of the slide-to-slide variance, the total synapsis efficiency was normalized for the number of molecules obtained from the reaction of Ku and X4L4 conducted on the same slide. Instead of using 'synapsis efficiency', 'normalized number of detected nonzero FRET molecules' was used to define the association frequency of the incoming dsDNA end contacting the immobilized dsDNA at specific region (end or middle) (Supplementary Figs. 1g and 4g). The molecules exhibiting at least one nonzero FRET portion within the FRET trajectory were counted to calculate the normalized number of detected nonzero FRET molecules.

All identified trajectories were then exported, and both donor and acceptor raw intensity data were smoothed by averaging the neighboring five data points. The data was further analyzed using programs written in MATLAB as described previously[13,44]. The FRET efficiency ($E_{FRET}$ or $E$) was calculated by Eq. (1)

$$E = \frac{I_A - D_{leakage} \times I_D}{\gamma \times I_D + I_A} \tag{1}$$

$I_A$: apparent acceptor intensity measured in the acceptor emission channel. $I_D$: donor intensity measured in the donor emission channel. $D_{leakage}$: leakage of the donor emission into the acceptor detection channel, defined as 8% in our system. $\gamma$: correction factor, defined as 1 in our system, which is related to the quantum yields of donor and acceptor, and detection efficiency of CCD cameras for the donor and acceptor. The background correction sometimes gave the calculated $E_{FRET}$ value out of the range between 0 and 1; therefore, the FRET efficiency ($E_{FRET}$) between −0.2 and 1.2 was then plotted in all the graphs.

All the molecules from each specific condition were combined and used for normalized $E_{FRET}$ histograms and further analysis. Histograms were generated after removing the photobleached portions of trajectories, and all the nonbleaching data points were included for the histograms.

**Grouping of synaptic events**. After removing the bleaching portions, the synapsis events without any transitions between low $E_{FRET}$ (<0.6) and high $E_{FRET}$ (≥0.6)

were automatically grouped based on their mean $E_{FRET}$ values over the entire event. The events with $E_{FRET} < 0.6$ were designated as **FS** complexes; the events with $E_{FRET} \geq 0.6$ were designated as **CS** complexes. Events with transitions between the two FRET states were manually truncated into several specific segments, which were further classified into different groups. The final segments in each group were used to calculate the ratio, the dwell time, and the $E_{FRET}$ distribution of different synaptic complexes.

**Synapsis time analysis**. The dwell time of the **FS** complex (Figs. 1c, 2c, and Supplementary Fig. 3c) was plotted based on the Cy3 signal of identified FRET pairs. This includes both no FRET interaction portions and FRET interaction portions of each synapsis event. The dwell time of the **CS** complex (Figs. 3e, 4d, and Supplementary Figs. 6e, f) was plotted based on the high $E_{FRET}$ ($E_{FRET} \geq 0.6$) of selected events. All the dwell time histograms and the corresponding exponential fits only correspond to the synapsis events with both beginning and end times within the detection time window.

**Quantification and statistical methods**. Gaussian fit of the $E_{FRET}$ distribution and exponential fit of the dwell time distribution were performed using OriginLab Origin 2019 and MATLAB R2018a, respectively. Unpaired, two-tailed $t$-test was performed using GraphPad Prism 7. All other data were compiled in Microsoft Excel.

**Reporting summary**. Further information on research design is available in the Nature Research Reporting Summary linked to this article.

## Data availability
The authors declare that the main data supporting the conclusions are provided in the files of source data1 and source data2. All data are available from the authors upon reasonable request.

## Code availability
The iSMS 2.01 software[42] was downloaded at http://inano.au.dk/about/research-groups/single-molecule-biophotonics-group-victoria-birkedal/software/.

MATLAB codes for further data analysis, including $E_{FRET}$, and dwell distribution analysis after trajectory extraction, were from Eli Rothenberg's lab.

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

## Acknowledgements

We thank all members of the Lieber lab for helpful discussions. B.Z. would like to thank Dr Nicholas R. Pannunzio, Christina Gerodimos, and Dr Raymond D. Mosteller for proofreading the paper, and Christina Gerodimos for providing the snap-tag free version of X4L4. We acknowledge the Cell and Tissue Imaging Core for use of the Nikon Eclipse Ti microscope, which receives support from the USC Norris Comprehensive Cancer Center P30 grant. We would also like to thank Dr. Ramunas Stanciauskas and Andrew Brumm from Nikon company for setting the TIRF microscopy, and Dan Vo from Dr Myron F. Goodman's lab at USC for helpful suggestions. The PAGE gel was imaged using the Typhoon machine in the Center of Excellence in NanoBiophysics at the University of Southern California. This work was supported by National Institutes of Health grants (GM118009, CA196671, CA100504 to M.R.L. and R01 GM108119 to E.R.), American Cancer Society Grant 130304-RSG-16-241-01-DMC (E.R.), and the V Foundation for Cancer Research Grant D2018-020 (E.R.).

## Author contributions

Conceptualization, methodology, and validation, B.Z. and M.R.L. Formal analysis and investigation, B.Z. and M.R.L. Writing–original draft, review, and editing, B.Z., E.R., and M.R.L. Resources, supervision, and project administration, M.R.L. Funding acquisition, M.R.L. G.W. provided DNA-PKcs and tested its activity. M.J.M. performed the auto-correlation analysis. D.A.R provided initial suggestions for the smFRET method development. All authors contributed to the revision of the paper and approved the final version.

## Additional information

**Competing interests:** The authors declare no competing interests.

