## [Peer Review File · Nature Communications]

Editorial Note #1: Parts of this Peer Review File have been redacted as indicated to maintain the confidentiality of unpublished data.

Editorial Note #2: In their second rebuttal, the authors respond to comments added by Reviewer #3 to an annotated manuscript file. The reviewer's comments have not been copied here from the annotated manuscript file.

Reviewers' comments:

Reviewer #1 (Remarks to the Author):

Zhao et al. present single molecule FRET data using purified proteins to investigate the mechanism of synapsis in DSB repair by non-homologous DNA end joining (NHEJ). The data reveal two phases in DNA end capture: transient pairing that is promoted by Ku and XRCC4-Lig4, and a more stable paired conformation that also requires XLF. PAXX stabilizes paired ends as well, but not as efficiently as XLF. This biphasic model mostly agrees with studies from the Loparo group using extracts of *X. laevis* eggs. Zhao et al report that DNA-PKcs is not required for end synapsis, which contradicts earlier data from biochemical assays and single molecule FRET data from the *Xenopus* system.

Overall, the data are of high quality and help shed light on end synapsis in NHEJ, a subject that is of general interest and that has provoked some controversy in the NHEJ literature. The following issues will need to be addressed:

1. A critical issue here is that the authors' reconstituted system does not yield the same requirement for DNA-PKcs as the Loparo group's work using *Xenopus* extracts. Particular emphasis needs to be placed on discussion of differences between the two systems. As it stands, the paper barely discusses this issue. My next several points directly address this issue:
2. Does protein concentration account for the lack of requirement for DNA-PKcs? There is likely much more X4L4 in the authors' reconstituted system than there is in *Xenopus*. Would the authors see a requirement for DNA-PKcs if X4L4 were to be titrated down? The authors test only one concentration of X4L4, 50 nM, in Figure 2a. The gel that contained lanes 1 and 2 needs to be shown.
3. Is DNA-PKcs active in the authors' assay system? The authors should verify that their protein has kinase activity in the buffer used for the single molecule assays. Does the protein show a monodispersed peak on a gel filtration column? Protein quality control is critical here.
4. Another possibility is that the authors' prep of DNA-PKcs is already autophosphorylated due to the presence of DNA during purification. DNA-PKcs autophosphorylation has been suggested to release it from the synaptic complex to allow end processing enzymes to access the ends. Pre-phosphorylated DNA-PKcs may not possess the end bridging activity that has been reported by others.
5. DNA-PKcs was only tested with Ku and X4L4. Does it have any effect in the presence of XLF or PAXX? In the *Xenopus* extract system, all of these NHEJ components would be present.
6. The gel shown in Figure 6 should be quantitated and presented as a graph showing ligation efficiency with error bars.
7. Do the dynamics of end synapsis change when overhangs are present to guide annealing? The authors could repeat a few key experiments with compatible vs. incompatible overhangs to compare with the blunt end data they have already shown.
8. The ends used here have 5' OH ends to block ligation. Adenylation of the 5' phosphate by the ligase may affect formation of the synaptic complex, and the 5' phosphate also affects Ku engagement. Discuss these issues at the minimum.
9. The authors open the abstract with the rhetorical claim that end synapsis is "the most central question about the repair of a double-strand DNA break" and later italicize the statement "The

importance of direct insight into the synapsis step cannot be overstated" in the Introduction. I suggest they tone these down.

10. On page 11-12 of the Discussion the authors write (regarding the proposed requirement for DNA-PKcs for end synapsis in mammals) "genetics tells us that this cannot be true because many components are not present in yeast and many invertebrates." This statement seems overly simplistic. The fact that yeast does not have DNA-PKcs does not preclude the idea that DNA-PKcs is required for end synapsis in mammalian cells. Other factors, such as the MRX complex, may substitute for DNA PKcs in DNA end synapsis.

Reviewer #2 (Remarks to the Author):

The present study by Zhao et al. extends the works by the Loparo, the Strick, and the Rothenberg labs that have used time-resolved SM assays (SM-FRET or magnetic tweezers) to study how small blunt-ended DNA molecules are brought together to facilitate ligation by core components of the NHEJ machinery.

Here Zhao et al. find that:

1) Ku + X4L4 are sufficient to form flexible synapsis which are mobile, side-by-side DNA interactions favorable to the accessibility of end processing enzymes but not for ligation.

Importantly, DNA-PKcs is not required for flexible synapsis

2) Transition of flexible synapsis to stable closed synaptic state is promoted by a few XLF dimers that trigger evolution of flexible synapsis into an end-to-end configuration favorable for ligation

3) PAXX also promotes transition of flexible synapsis to stable closed synaptic state but less efficiently and differently than XLF

Overall it is a well-documented and impressive study that has the merit of integrating, extending and refining all the previous SM studies that reconstituted minimal NHEJ in vitro. Zhao et al. have developed a powerful methodology and a conceptual frame of work that can now be used to study NHEJ in more detail (such as influence of DNA end structure, end processing factors, post-translational modifications). For these reasons, publication of this work in Nature Communications is appropriate.

One important finding of the study is that DNA-PKcs is not required in such assays, which is consistent with the genetic data and many biochemical studies. However, this finding clashes with the conclusion of the Lopara and Strick labs studies where DNA-PKcs appears to be involved.

This reviewer would like to suggest addressing this point further in order to try to resolve this discrepancy. Possibilities are:

1) Given that the interactions promoted by DNA-PK holoenzyme are weak and transient they must be more difficult to detect and monitor. How do the technical procedures to mix and quickly analyze the samples vary in the different labs? Could this be a factor?

2) There is clearly a difference between Fig1d (Ku, X4L4) and Fig2b (Ku, DNA-PKcs, X4L4). When DNA-PKcs is added to Ku + X4L4, a shoulder to higher EFRET is clearly visible in the distribution. Isn't this telling us something about DNA-PKcs?

3) DNA-PKcs is added at 12.5 nM and Ku at 25 nM (dimers?). What is the impact of adding 25 nM or higher concentrations of DNA-PKcs?

4) Are there two sub-pathways to promote end-to-end alignment (Ku+X4L4+XLF, and DNA-PK holoenzyme+ X4L4+PAXX). Experiments with Ku + DNA-PKcs + X4L4 + PAXX could be informative and this combination of factors is missing in the study.

5) Potential effects of post-translation modifications of recombinant proteins

Minor comments:

. Further discuss the role of X4, XLF and L4 filaments? Do these SM assays with small DNA substrates actually reconstitute "physiological" NHEJ? The opinion of this reviewer is that the SM assays used here and by the Loparo and Strick labs are very informative to study the chemistry at DNA extremities but do not reconstitute repair of broken chromosome by NHEJ.

. If DNA-PKcs is not required for DNA end synopsis, discuss what would be its physiological roles.

Reviewer #3 (Remarks to the Author):

In this work, Lieber and coworkers use single molecule fluorescence to examine synopsis mediated by NHEJ proteins. Overall the work is potentially interesting; however, there are serious flaws in the data analysis that make it difficult to assess the significance and make it inappropriate for publication at this time. In addition, much of the data, such as dwell time plots, used to determine lifetimes are not shown and so it is difficult to evaluate. Finally, the data analysis requires a more robust statistical analysis. I outline the major issues below. With a careful analysis, this work may prove to be very interesting.

1) For the FRET distributions, the authors state that they determine the average FRET efficiency based on a Gaussian fit to the data; however, no fits are shown on any of the data. A greater problem is that none of the distributions shown in the figures appears to be a single Gaussian, indicating multiple peaks.

Fig 1d shows a long tail at high FRET.

Fig. 2b shows two clear peaks (~.1 & .3) and a tail, suggesting three states.

Fig.3d shows a peak at ~.1 with a tail and a peak at ~.8 (identified in Fig.)

Fig. 4b shows 2 low FRET peaks (only one identified) and one high FRET (identified)

Fig. 4e shows at least 3 clear peaks.

The authors did not characterize or discuss any of these peaks. For example, on page 5, they state "DNA-PKcs also does not have a significant effect on the FRET efficiency (EFRET) distribution of the synaptic complex formed by Ku and X4L4 (Fig. 1d, 2b)."; however, it looks like it has a dramatic effect. There is a new significant peak at FRET ~.3. Interestingly, in Fig. 3d when DNA-PKcs is added, it is the .3 peak that disappears and the 0.1 peak remains.

A close analysis of the distributions and how they change upon adding additional proteins may give better insight into the mechanism than simply defining what appears to be an arbitrary cutoff of a FRET of 0.6 to define different states.

2) There also appear to be problems with the identification of transitions in the individual FRET traces. The authors use HaMMY to determine transitions. This procedure, like most procedures, depends on user input of thresholds for transitions. It appears that in many cases, noise in the signal is being identified as transitions. For example, in Fig. 2d, HaMMY is clearly identifying noise as transitions. At the end of the trace (as well as other places), HaMMY identifies transitions where

there is no evidence of anticorrelation in the donor and acceptor intensity traces, as expected for FRET. Similar effects can also be seen in Fig 1b. There is no statistical test, such as a t-test to confirm transitions. Perhaps the authors should consider other methods for identifying transition or at the very least they must present a statistical analysis to confirm their significance. It is clear from the traces that HaMMY is over identifying transitions, which is biasing the results.

3) The over identification of transitions by HaMMY results in dwell time plots that are biased to short times, and it leads to the potentially erroneous conclusion that data in Fig 1 and 2 show no differences in lifetime. Comparison of Figs. 1b and 2d show that the transitions between FRET .1 and .3 appear to be more frequent in 1b than 2d (1-2s vs 5-10s), suggesting that the 0.3 FRET state is longer lived and more populated in the presence of DNA-PKcs. This suggestion is consistent with the appearance of the peak at ~ 0.3 in the FRET distribution in Fig. 2b

4) The dwell time data in Fig. 1d look like they may be better fit to a two-step mechanism, because the first bin has a lower occupancy than the second bin. This possibility could be better differentiated with slightly finer binning of the data.

5) It is unclear how the "rates" (rate constants?) shown in Figs. 1f, 2e, and 4d are determined or . The Methods indicate that $n > 20$ events was used for rates. It is unclear if events mean transitions or traces, and this number events is quite small for determination of a rate constant from a dwell time plot. Also, the source of the error bar is not stated. The authors need to show the dwell time plots and the fits to the data for the reader to judge the comparison.

6) Fig. 2c show a survival probability plot but it is unclear how the data were evaluated to generate this plot. They provide a reference to another paper; however, they do not appear to have the same issue of too few transitions in cited reference. The authors should more clearly explain their analysis. Perhaps the authors could take longer movies to circumvent this problem, since the donor is in solution and photobleaching should be less of a problem.

7) The discussion of synopsis efficiency on page 6 is hard to follow. The word synopsis appears to have different meaning throughout the paper (those that ligate, those that come together with any FRET, only those that come together with high FRET), which is confusing.

8) On page 6, the authors state "A higher synopsis efficiency is observed for the end labeled configuration, suggesting that the binding of the two duplexes occurs preferentially at the DNA ends." This result does not make sense. The complexes should be the same, unless the dyes are affecting with complex formation. How is efficiency calculated? What is the error?

Responses to Reviewer Comments (author responses in BLUE)

The authors would like to thank the three Reviewers and the Editor for their time in evaluating our study and for making useful suggestions. We believe that we have addressed all of the comments below, but we are open to any additional ones.

Reviewer #1

Zhao *et al.* present single molecule FRET data using purified proteins to investigate the mechanism of synapsis in DSB repair by non-homologous DNA end joining (NHEJ). The data reveal two phases in DNA end capture: transient pairing that is promoted by Ku and XRCC4-Lig4, and a more stable paired conformation that also requires XLF. PAXX stabilizes paired ends as well, but not as efficiently as XLF. This biphasic model mostly agrees with studies from the Loparo group using extracts of *X. laevis* eggs. Zhao *et al.* report that DNA-PKcs is not required for end synapsis, which contradicts earlier data from biochemical assays and single molecule FRET data from the Xenopus system.

Overall, the data are of high quality and help shed light on end synapsis in NHEJ, a subject that is of general interest and that has provoked some controversy in the NHEJ literature. The following issues will need to be addressed:

1. A critical issue here is that the authors' reconstituted system does not yield the same requirement for DNA-PKcs as the Loparo group's work using *Xenopus* extracts. Particular emphasis needs to be placed on discussion of differences between the two systems. As it stands, the paper barely discusses this issue.

In our system, DNA-PKcs is not required for the Flexible Synapsis (**FS**) or the Close Synapsis (**CS**) states. The clearest and strongest observation that DNA-PKcs is not required for synapsis is that we observe efficient covalent ligation without any DNA-PKcs in the ensemble (bulk solution) studies not only here (Fig. 6a), but also in earlier studies (PMID: 27703001, PMID: 27705800). Ligation would be impossible without synapsis. These results provide a clear answer for purified systems.

For crude extract systems, as in the Loparo study, numerous other components in the *Xenopus laevis* egg extracts beyond simply the NHEJ proteins might bind nonspecifically (e.g., histones, HMG proteins) or specifically to the dsDNA ends. DNA-PKcs might phosphorylate these or physically displace them. It is nearly impossible for us to list all of the possible interactions in a crude extract, most of which are not physiologically interesting. Nevertheless, in the original submission (old Supplementary Fig. 1d legend), we did state that "although we do not observe synapsis by Ku and DNA-PKcs, we cannot completely exclude the possibility that Ku plus DNA-PKcs can mediate the 'long-range' synapsis observed by the Loparo group." On this point regarding Ku + DNA-PKcs (without other proteins), we agree with the reviewer that this deserves a more detailed study of its own. For this reason, we have removed from the paper discussion of whether Ku + DNA-PKcs can mediate synapsis in the absence of other proteins, including in the absence of X4L4.

We thank the reviewer for suggesting additional discussion of this point, which we had limited in the original submission due to length limitations. But we have now included a few additional sentences on page 14. Of note, the Loparo group used a two-laser system that might detect looser or longer range interactions, or might detect nonspecific interactions that are abundant in crude extracts. Importantly, we found that the PCA+PCD oxygen scavenger system used in the imaging buffer almost completely inhibits the kinase activity of DNA-PKcs. Importantly, the Loparo study used PCA+PCD also. Therefore, their result that DNA-PKcs kinase activity is required for the 'short-range synapsis' in their paper is now subject to uncertainty. This uncertainty applies to both their Molecular Cell 2016 and their Nature Structure & Molecular Biology 2018 studies. The killing of DNA-PKcs kinase activity in PCA+PCD systems that we have now identified (see below) makes it even more important that our current submission be published. Otherwise, others will be misled by the Loparo study using dead DNA-PKcs. [We have repeated all of our DNA-PKcs studies with a different oxygen scavenger system in which DNA-PKcs activity is retained (see below).]

My next several points directly address this issue:

2. Does protein concentration account for the lack of requirement for DNA-PKcs? There is likely much more X4L4 in the authors' reconstituted system than there is in *Xenopus*. Would the authors see a requirement for DNA-PKcs if X4L4 were to be titrated down? The authors test only one concentration of X4L4, 50 nM, in Figure 2a. The gel that contained lanes 1 and 2 needs to be shown.

Thank you for the suggestions about protein concentration. The Loparo group does not appear to specify the concentrations of NHEJ proteins in the *Xenopus* extracts used in their papers. Based on their rescue

experiments, X4L4 at 'tens of nM' concentration (50 nM) was also supplemented to their X4-depleted extracts (PMID: 26990988). Since Ku and X4L4 at these concentrations can mediate efficient synapsis, we did not titrate the X4L4 concentration. We speculate that less synapsis would be observed when using less X4L4 in the solution. In the submitted paper, we found Ku and X4L4 can mediate efficient synapsis, and DNA-PKcs does not have a significant effect on the synapsis mediated by Ku and X4L4 (Fig. 2 and Supplementary Fig. 3). Moreover, we found DNA-PKcs is not required for the transition from **FS** to **CS** as tested not only in our previous PCA+PCD system, but also now in the glucose-gloxy system (DNA-PKcs kinase activity is retained). Although DNA-PKcs in solution does not have a significant effect on the **FS** complex, we found DNA-PKcs (at the same concentration as used in the Ku+X4L4+DNA-PKcs experiment) actually reduces the formation of **CS** complex as tested in the imaging buffer with glucose-gloxy oxygen scavenger system (Supplementary Fig. 7). Since the synapsis near the dsDNA ends is monitored by FRET signal and completely zero FRET synapsis (such as the 'long-range synapsis' as observed by the Loparo group) could not be observed using our system, we cannot completely exclude the possibility that two synapsis pathways simultaneously exist for flexible synapsis (**FS**). But we are sure that DNA-PKcs is not needed for **CS**, which is ready for ligation and exhibits a high FRET. Further experiments using a two-laser system as Dr. Loparo did might provide more information about this issue. We have added this information into the Discussion (page 14).

We have added the protein concentration labels onto the gels, in addition to being in the legend and Methods (Fig. 6a and Supplementary Figure 9). Since 40 nM dsDNA in the ensemble (bulk solution) reaction vs 10nM in the single molecule assay was used, the protein concentration was doubled in the ensemble reaction. The protein concentrations for the ensemble ligation reactions are: 50 nM Ku, 25 nM DNA-PKcs, 100 nM X4L4, 100 nM XLF, and 100 nM PAXX, and are specified in the legend and Methods.

3. Is DNA-PKcs active in the authors' assay system? The authors should verify that their protein has kinase activity in the buffer used for the single molecule assays. Does the protein show a monodispersed peak on a gel filtration column? Protein quality control is critical here.

Our endogenous DNA-PKcs was purified from HeLa cells to homogeneity using a series of chromatography steps and the last chromatography step was a size exclusion column (Superdex 200 Increase 10/300 GL). The chromatogram of the size exclusion column always showed a monodispersed peak at the fraction corresponding to ~470 kDa (provided for reviewers-only below).

DNA-PKcs kinase activity is quite reduced by the PCA+PCD components in the buffer, and this is an important point that we now highlight in the paper, since other labs (including the Loparo lab) that have studied DNA-PKcs by smFRET failed to point out this reduction. For this reason, we have now repeated all studies involving DNA-PKcs using the other major oxygen scavenger system, which is the glucose-gloxy (glucose oxidase and catalase) system. We documented that DNA-PKcs kinase activity is robust in the glucose-gloxy system (Supplementary Fig. 2b). As we knew from the Figure 6a ensemble studies, DNA-PKcs is not required in our smFRET system and does not stimulate synapsis by Ku + X4L4 (new Figure 2). We thank the Reviewer for suggesting that we check the effect of PCA+PCD on DNA-PKcs activity. We find similar results on **FS** and **CS** synaptic complexes using both the PCA+PCD and the glucose-gloxy oxygen scavenger systems (Supplementary Figs. 3 and 7).

4. Another possibility is that the authors' prep of DNA-PKcs is already autophosphorylated due to the presence of DNA during purification. DNA-PKcs autophosphorylation has been suggested to release it from the synaptic complex to allow end processing enzymes to access the ends. Pre-phosphorylated DNA-PKcs may not possess the end bridging activity that has been reported by others.

Our DNA-PKcs is purified from log phase HeLa cells. Although some fractions of DNA-PKcs within the cells were already partially phosphorylated, our purification strategy enriches only the active form of DNA-PKcs, which is the species that can be further phosphorylated. Even though our purified DNA-PKcs, which we believe is structurally and functionally highly homogeneous, might contain a very small fraction of partially phosphorylated DNA-PKcs at the end of the purification, the majority (or almost all) of our DNA-PKcs can be activated by an addition of dsDNA (Supplementary Fig.2). Therefore, our DNA-PKcs is not fully phosphorylated and does have substantial capability to be further phosphorylated upon DNA binding (PMID: 15574326 and PMID: 18722175). Also, our DNA-PKcs has strong DNA end binding activity (PMID: 9742108).

5. DNA-PKcs was only tested with Ku and X4L4. Does it have any effect in the presence of XLF or PAXX? In the Xenopus extract system, all of these NHEJ components would be present.

We have tested the effect of DNA-PKcs in the presence of XLF or PAXX using the imaging buffer containing glucose + glyoxy oxygen scavenger system, and added the information into Supplementary Fig. 7. Briefly, DNA-PKcs slightly inhibits the formation of **CS** complex (Supplementary Fig. 7c). This might be because the binding of DNA-PKcs at the dsDNA ends but not quick dissociation within some synaptic complexes causes some steric hindrance, which blocks the accessibility of the dsDNA ends for XLF and/or PAXX.

6. The gel shown in Figure 6 should be quantitated and presented as a graph showing ligation efficiency with error bars.

We thank the Reviewer for the suggestion. We had provided a quantified graph shown in the original submission Supplementary Figure 6b. We have moved it to Fig. 6a.

7. Do the dynamics of end synopsis change when overhangs are present to guide annealing? The authors could repeat a few key experiments with compatible vs. incompatible overhangs to compare with the blunt end data they have already shown.

We thank the Reviewer for the helpful suggestion. We have done and are continuing similar experiments using compatible and partially compatible overhangs (where the compatibility is recessed to varying extents). The major finding is that any base pairing between the ends makes synopsis much easier. Therefore, the most difficult and most interesting case is the one described here for blunt ends. As the Reviewer knows, once one begins to use compatible or partially compatible DNA ends (especially with varying extents of recess), then there are many thousands of variations. We anticipate that it will be a very long time before we arrive at any general conclusions, but the basic one, as stated above, is that blunt ends provide the clearest view of the essential elements for synopsis, which is the title and focus of this study. Note that we summarized all of this unpublished work of ours in the original submission Discussion (page 15). We state that 'once the ends can transiently pair via microhomology at the ends, Ku and X4L4 are sufficient to promote the close synaptic (**CS**) complex formation without XLF'.

8. The ends used here have 5' OH ends to block ligation. Adenylation of the 5' phosphate by the ligase may affect formation of the synaptic complex, and the 5' phosphate also affects Ku engagement. Discuss these issues at the minimum.

We appreciate the Reviewer's concern. But we note that charged ligase 4 requires that the DNA end already have a 5' phosphate, and it adds an AMP to create a phosphoanhydride. Ligases cannot adenylate a 5'-OH. Apart from this, we have added text on page 17 to mention awareness of the status of the 5'-OH versus 5'-P, as the Reviewer suggested.

9. The authors open the abstract with the rhetorical claim that end synopsis is "the most central question about the repair of a double-strand DNA break" and later italicize the statement "The importance of direct insight into the synopsis step cannot be overstated" in the Introduction. I suggest they tone these down.

We have modified our text accordingly (page 2 and 3).

10. On page 11-12 of the Discussion the authors write (regarding the proposed requirement for DNA-PKcs for end synopsis in mammals) "genetics tells us that this cannot be true because many components are not present in yeast and many invertebrates." This statement seems overly simplistic. The fact that yeast does not have DNA-PKcs does not preclude the idea that DNA-PKcs is required for end synopsis in mammalian cells. Other factors, such as the MRX complex, may substitute for DNA PKcs in DNA end synopsis.

We have modified our text accordingly (page 13).

[REDACTED]

Reviewer #2

The present study by Zhao *et al.* extends the works by the Loparo, the Strick, and the Rothenberg labs that have used time-resolved SM assays (SM-FRET or magnetic tweezers) to study how small blunt-ended DNA molecules are brought together to facilitate ligation by core components of the NHEJ machinery.

Here Zhao *et al.* find that:

1) Ku + X4L4 are sufficient to form flexible synapsis which are mobile, side-by-side DNA interactions favorable to the accessibility of end processing enzymes but not for ligation.

Importantly, DNA-PKcs is not required for flexible synapsis

2) Transition of flexible synapsis to stable closed synaptic state is promoted by a few XLF dimers that trigger evolution of flexible synapsis into an end-to-end configuration favorable for ligation

3) PAXX also promotes transition of flexible synapsis to stable closed synaptic state but less efficiently and differently than XLF

Overall it is a well-documented and impressive study that has the merit of integrating, extending and refining all the previous SM studies that reconstituted minimal NHEJ *in vitro*. Zhao *et al.* have developed a powerful methodology and a conceptual frame of work that can now be used to study NHEJ in more detail (such as influence of DNA end structure, end processing factors, post-translational modifications). For these reasons, publication of this work in Nature Communications is appropriate.

One important finding of the study is that DNA-PKcs is not required in such assays, which is consistent with the genetic data and many biochemical studies. However, this finding clashes with the conclusion of the Lopara and Strick labs studies where DNA-PKcs appears to be involved.

We thank the Reviewer for highlighting one of our key findings. The difference between our study and the two earlier ones may be because of the following reasons:

For the *Xenopus* extract, other components in the *Xenopus laevis* egg extracts beyond just the NHEJ proteins might competitively bind to the dsDNA end, which could affect the synapsis process mediated by NHEJ factors. DNA-PKcs may be required to release these bound non-NHEJ competitors [which may be quite nonspecific, such as free nonnucleosomal histones and HMG proteins]. Moreover, although we did not observe any evidence of DNA-PKcs on the synapsis mediated by Ku and X4L4 in our system, we do not completely exclude the possibility that Ku and DNA-PKcs may mediate a very 'long-range' synapsis (estimated to be >12.5 nm from our interpretation of the Loparo group's data). But we clearly found that DNA-PKcs is not required for flexible synapsis (**FS**) mediated by Ku and X4L4 and also not required for the close synapsis (**CS**) mediated by Ku + X4L4 + XLF or PAXX. We have addressed these possibilities in the Discussion (page 14). Moreover, we found DNA-PKcs kinase activity was severely reduced by the PCA+PCD oxygen scavenger system in the smFRET imaging buffer. The same PCA+PCD system was used in the Loparo study, which severely complicates their conclusions. They proposed DNA-PKcs kinase activity is required for the transition from the 'long-range synapsis' to the 'short-range synapsis'. The results about DNA-PKcs and its kinase activity in the two Loparo papers are therefore likely questionable.

For the Strick system, two dsDNA molecules are tethered together using a dsDNA bridge, to which NHEJ proteins such as XLF and X4 could potentially also bind in a manner that may complicate the studies and reduce the effective free concentration of some NHEJ proteins over others. The dsDNA bridge and protein binding onto the experimental scaffold (the bridge) might influence synapsis. In the Strick study, they almost did not detect the ligation products when DNA-PKcs was omitted from their system. The requirement of DNA-PKcs for ligation is inconsistent with many previous biochemical studies (PMID: 27703001, PMID: 27705800), which suggests their reconstituted system for NHEJ was not representative.

This reviewer would like to suggest addressing this point further in order to try to resolve this discrepancy. Possibilities are:

1) Given that the interactions promoted by DNA-PK holoenzyme are weak and transient they must be more difficult to detect and monitor. How do the technical procedures to mix and quickly analyze the samples vary in the different labs? Could this be a factor?

The proteins were added in the following order within a one minute assembly period: 25 nM Ku, 50 nM X4L4, 12.5 nM DNA-PKcs, 50 nM XLF, and 50 nM PAXX (proteins for specific experiments were stated in the text and legends). The incoming dsDNA was added last. Once the dsDNA was added, the reaction mixture was quickly mixed and immediately injected into the chamber, and the images were immediately captured (each step within 15 sec). If the interactions are dynamic and reversible, then the mixing of the reaction components would not be an issue for detection, because the association and dissociation of the two dsDNAs mediated by NHEJ proteins can occur throughout the detection period.

Regarding the very fast dynamic and transient interactions, the interactions can be detected as long as the capture rate of the detector (CCD camera) is faster than the dissociation rate of the synaptic complex. The duration of the long-range synaptic complex (Ku and DNA-PKcs are necessary for this kind of complex) described in Dr. Loparo's paper is at the several-second time-scale. Though using DNA substrate that was already linked by a dsDNA bridge, the Strick study found the lifetime of the synaptic complex formed by Ku and DNA-PKcs to be at the 100 ms time scale. We used a 50 ms exposure time for the CCD camera detection. We would not have missed any transient interaction unless the lifetime of the synaptic complex is < 50 ms. We have now made certain that this is all clear in the text. We thank the Reviewer for the suggestions.

2) There is clearly a difference between Fig1d (Ku, X4L4) and Fig2b (Ku, DNA-PKcs, X4L4). When DNA-PKcs is added to Ku + X4L4, a shoulder to higher EFRET is clearly visible in the distribution. Isn't this telling us something about DNA-PKcs?

We understand the Reviewer's concern here. Primarily, we observed two different kinds of synaptic complexes - **FS** and **CS** complexes. Within the **CS** complex, the two dsDNAs are aligned in an end-to-end configuration, and the **CS** complex is nearly homogeneous (as reflected in the E_{FRET} distribution). In contrast, the two dsDNAs are only laterally aligned within the **FS** complex. The distance between the two dsDNA ends varies within different **FS** complexes, which results in the broad distribution of the E_{FRET} . In other words, the dsDNA end can contact the other dsDNA molecule along its duplex length at many positions. Therefore, the set of **FS** complexes has many degrees of freedom, even though this entire set has a lateral parallel alignment (i.e., side-by-side). The appearance of a small shoulder toward higher E_{FRET} when DNA-PKcs is included in the system (previous Fig.2 in the initial submission) might indicate a slight accumulation of synaptic complex, within which the two ends of the side-by-side aligned dsDNA are relatively close. The binding of DNA-PKcs at the dsDNA ends might be responsible for this small shoulder. But DNA-PKcs would not cause a large effect on the structure of the synaptic complex formed by Ku plus X4L4. It could not drive the **FS** complex (lateral alignment of the two dsDNAs) to form a **CS** complex (end-to-end configuration). Moreover, we tested the kinase activity in the smFRET imaging buffer as Reviewer 1 suggested and found that DNA-PKcs kinase activity was severely reduced in the previous imaging buffer containing the PCA + PCD oxygen scavenger system. We have repeated all the experiments involving DNA-PKcs using the imaging buffer containing the glucose + gloxy oxygen scavenger system, in which DNA-PKcs retains its kinase activity. The E_{FRET} distributions of **FS** complexes formed with and without DNA-PKcs in the reaction solution show no substantial difference (Fig. 2b and Supplementary Fig. 3a).

3) DNA-PKcs is added at 12.5 nM and Ku at 25 nM (dimers?). What is the impact of adding 25 nM or higher concentrations of DNA-PKcs?

25 nM Ku stands for the concentration of Ku70/Ku80 heterodimer. We have not tried 25 nM or higher concentrations of DNA-PKcs. But based on our previous results (PMID: 25941401), higher concentrations of DNA-PKcs (which was purified and provided by our lab) cause substantial aggregation of the DNA in solution.

4) Are there two sub-pathways to promote end-to-end alignment (Ku+X4L4+XLF, and DNA-PK holoenzyme+X4L4+PAXX). Experiments with Ku + DNA-PKcs + X4L4 + PAXX could be informative and this combination of factors is missing in the study.

We thank the Reviewer for the suggestion. Here we found DNA-PKcs is not required for either the **FS** and **CS**. Since the synapsis near the dsDNA ends is monitored by FRET signal and complete zero FRET synapsis (such as the 'long-range synapsis' as observed by the Loparo group) could not be observed using our system, we cannot completely exclude the possibility that two synapsis pathways (the Ku + X4L4 pathway we describe here as well as a Ku + DNA-PKcs pathway) simultaneously exist for flexible synapsis (**FS**). But we are certain that DNA-PKcs is not needed for **CS**, which is a configuration ready for ligation and exhibits high FRET. Further experiments using a two-laser system as the Loparo group did might provide more information about this issue. We have added this information to the Discussion (page 14).

As the Reviewer suggested here, we have also tested the synopsis mediated by Ku+DNA-PKcs+X4L4+PAXX. The corresponding results have been added into Supplementary Fig. 7. Briefly, instead of promoting the formation of **CS** complex, DNA-PKcs presence in the Ku+X4L4+PAXX reaction system slightly inhibits the formation of **CS** complex. This is consistent with the ensemble experiments here (Fig. 6a) and previously (PMID: 27703001, PMID: 27705800).

5) Potential effects of post-translation modifications of recombinant proteins

We appreciate the Reviewer's suggestions. We have done mass spec on some NHEJ proteins previously, and we will continue to be vigilant about any possibility of such effects in the future.

Minor comments:

Further discuss the role of X4, XLF and L4 filaments? Do these SM assays with small DNA substrates actually reconstitute "physiological" NHEJ? The opinion of this reviewer is that the SM assays used here and by the Loparo and Strick labs are very informative to study the chemistry at DNA extremities but do not reconstitute repair of broken chromosome by NHEJ.

We thank the Reviewer for their suggestions. The Reviewer pointed out a good scientific question for the NHEJ field. Using naked dsDNA in our system, we found XLF plays an important role at the dsDNA end, but not through the filament structure to mediate synopsis. As for the chromatinized DNA substrates, filaments of XLF could be relevant. Another long-term project in our lab aims to reconstitute the repair of chromatinized DNA. We addressed the chromatinized DNA issue in our Discussion (page 17).

If DNA-PKcs is not required for DNA end synopsis, discuss what would be its physiological roles.

One of the major physiological roles of DNA-PKcs is to interact with and activate the endonuclease activity of Artemis at DNA ends. Therefore, we speculate that DNA-PKcs plus Artemis would be important and necessary for incompatible DNA end repair, but not for synopsis of the two blunt dsDNAs. Since DNA-PKcs has little effect on the synopsis mediated by Ku plus X4L4 discovered here, the flexibility of the DNA ends held by Ku and X4L4 in the **FS** may facilitate nucleolytic end processing by DNA-PKcs-activated Artemis. We have addressed the physiological roles of DNA-PKcs in the Discussion (page 14), given the suggestion of the Reviewer.

Reviewer #3

In this work, Lieber and coworkers use single molecule fluorescence to examine synopsis mediated by NHEJ proteins. Overall the work is potentially interesting; however, there are serious flaws in the data analysis that make it difficult to assess the significance and make it inappropriate for publication at this time. In addition, much of the data, such as dwell time plots, used to determine lifetimes are not shown and so it is difficult to evaluate. Finally, the data analysis requires a more robust statistical analysis. I outline the major issues below. With a careful analysis, this work may prove to be very interesting.

1) For the FRET distributions, the authors state that they determine the average FRET efficiency based on a Gaussian fit to the data; however, no fits are shown on any of the data. A greater problem is that none of the distributions shown in the figures appears to be a single Gaussian, indicating multiple peaks.

Fig 1d shows a long tail at high FRET.

Fig. 2b shows two clear peaks (~.1 & .3) and a tail, suggesting three states.

Fig.3d shows a peak at ~.1 with a tail and a peak at ~.8 (identified in Fig.)

Fig. 4b shows 2 low FRET peaks (only one identified) and one high FRET (identified)

Fig. 4e shows at least 3 clear peaks.

The authors did not characterize or discuss any of these peaks. For example, on page 5, they state "DNA-PKcs also does not have a significant effect on the FRET efficiency (EFRET) distribution of the synaptic complex formed by Ku and X4L4 (Fig. 1d, 2b)."; however, it looks like it has a dramatic effect. There is a new significant peak at FRET ~.3. Interestingly, in Fig. 3d when DNA-PKcs is added, it is the .3 peak that disappears and the 0.1 peak remains.

A close analysis of the distributions and how they change upon adding additional proteins may give better insight into the mechanism than simply defining what appears to be an arbitrary cutoff of a FRET of 0.6 to define different states.

We understand the Reviewer's concerns. The reviewer is correct that the FRET distributions are not single Gaussian. We have re-plotted and re-fitted our E_{FRET} distribution. The mixture of Gaussian fits and corresponding Gaussian peaks are now shown on the graphs.

The E_{FRET} values shown on the graphs were the fitted values of the main (highest) peaks of the distributions. We have modified our text to make the statements related to these fitted E_{FRET} values clearer.

The Reviewer is also correct that there are tails and small shoulders on some distributions. To simplify the description of the synopsis process in the manuscript, we focused on two primary states (represented by the two main peaks). But those tails and shoulders would not affect the description of the synaptic structures (represented by the two primary states) for the following reasons:

We observed two different kinds of synaptic complexes -- **FS** and **CS** complexes. The E_{FRET} peak of the **CS** complex is the same as the pre-ligated positive control of the two dsDNA substrates, as shown in Supplementary Fig. 4f. This kind of **CS** complex is relatively homogeneous. Within the **CS** complex, the two dsDNAs are aligned in end-to-end configuration (i.e. in-line close contact) (Fig.6b), and are at the same distance as a ligatable nick control. Based on the E_{FRET} distribution of the positive control (the lowest E_{FRET} boundary is ~ 0.6) and the distribution of the **CS** complex, E_{FRET} 0.6 was selected as the cutoff to identify the end-to-end configured **CS** complex.

Within the **FS** complex, the two dsDNAs are in parallel lateral alignment (i.e. side-by-side) (Fig. 6b). Therefore, one dsDNA end can contact any of multiple positions of the other dsDNA duplex within the **FS** complex. Within some of the **FS** complexes, the dsDNA can move or slide along one another. The distance between the two dsDNA ends varies within different **FS** complexes, which results in a broad distribution of the E_{FRET} . Therefore, the **FS** complexes include numerous intermediate structures. The intermediate structures might be interesting, but it is difficult to accurately solve their structures merely by the FRET method. We will focus on those intermediate structures later by using cryo-EM. Here, we simplified the description of this **FS** state by only focusing on the main (highest) peak. We have modified our text to make our statements clearer (page 7).

In the manuscript here, we found Ku and X4L4 mainly mediate the formation of the **FS** complex represented by the main peak with the small E_{FRET} value shown on Figure 1d, and DNA-PKcs does not have a substantial effect on the formation of the **FS** complex mediated by Ku plus X4L4 (previous Fig. 2b). The appearance of a small shoulder to higher E_{FRET} ($E: \sim 0.3$) in the old Figure 2 when DNA-PKcs was included in the system might indicate a slight accumulation of the synaptic complex, within which the two *ends* of the side-by-side aligned dsDNA are relatively closer. The binding of DNA-PKcs at the dsDNA ends might be responsible for this small shoulder. Nevertheless, DNA-PKcs does not have a large effect on the structure of the synaptic complex formed by Ku plus X4L4. The two dsDNA within the **FS** complex are still laterally aligned. DNA-PKcs was not able to drive the **FS** complex (lateral alignment of the two dsDNAs) to form a **CS** complex (end-to-end configuration). Moreover, we tested the DNA-PKcs kinase activity in the imaging buffer with the PCA + PCD oxygen scavenger system as Reviewer 1 suggested, and found the kinase activity was severely reduced by the PCA + PCD. We repeated all the experiments involving DNA-PKcs using another oxygen scavenger system, namely, glucose + gloxy (glucose oxidase and catalase), in which the DNA-PKcs kinase activity is retained and robust (Supplementary Fig. 2). We found similar results for synopsis mediated by Ku and X4L4 as described on page 7 when either the PCA + PCD system or the glucose + gloxy system was used in the solution. Also, we found that DNA-PKcs in solution does not have a large effect on the E_{FRET} distribution of synaptic complex formed by Ku and X4L4 using the glucose + gloxy oxygen scavenger system (new Fig. 2 and Supplementary Fig. 3).

For the Fig. 3d, there is no DNA-PKcs in the reaction system. Ku, X4L4, and XLF were included in the system. Here we found XLF could drive the **FS** complex to form the **CS** complex.

2) There also appear to be problems with the identification of transitions in the individual FRET traces. The authors use HaMMY to determine transitions. This procedure, like most procedures, depends on user input of thresholds for transitions. It appears that in many cases, noise in the signal is being identified as transitions. For example, in Fig. 2d, HaMMY is clearly identifying noise as transitions. At the end of the trace (as well as other places), HaMMY identifies transitions where there is no evidence of anticorrelation in the donor and acceptor intensity traces, as expected for FRET. Similar effects can also be seen in Fig 1b. There is no statistical test, such as a t-test to confirm transitions. Perhaps the authors should consider other methods for identifying transition or at the very least they must present a statistical analysis to confirm their significance. It is clear from the traces that HaMMY is over identifying transitions, which is biasing the results.

We thank the reviewer for the suggestions. We have smoothed the traces and re-done the HMM analysis using the HaMMY software (PMID: 16766620)(Fig. 1e and Supplementary Fig. 3d,e). The output report files from HaMMY were then subjected to the transition density plot (TDP) software (PMID: 16766620). The transition rates were then extracted from the TDP software. smFRET works on a different distribution than a t-test distribution;

therefore, a t-test is not used here. Moreover, HaMMY-TDP combination gives the statistical equivalent to the t-test (PMID: 16766620).

To further confirm that there is no significant difference between FRET trajectories from synapsis mediated by Ku and X4L4 in the presence and absence of DNA-PKcs, a 1-D autocorrelation function was used to process the dynamic trajectories obtained using the glucose + gloxy oxygen system (Supplementary Fig. 3g). The average autocorrelation values were plotted against time to visualize the difference between the dynamics of the trajectories from experiments with and without DNA-PKcs. The fitting of the autocorrelation curves to bi-exponential decays does not show any significant difference between the parameters using an unpaired t-test (Supplementary Fig. 3g).

3) The over identification of transitions by HaMMY results in dwell time plots that are biased to short times, and it leads to the potentially erroneous conclusion that data in Fig 1 and 2 show no differences in lifetime. Comparison of Figs. 1b and 2d show that the transitions between FRET .1 and .3 appear to be more frequent in 1b than 2d (1-2s vs 5-10s), suggesting that the 0.3 FRET state is longer lived and more populated in the presence of DNA-PKcs. This suggestion is consistent with the appearance of the peak at ~0.3 in the FRET distribution in Fig. 2b. We understand the Reviewer's concerns. Since the DNA-PKcs loses its kinase activity in the buffer containing PCA + PCD, we repeated all the experiments involving DNA-PKcs using the glucose + gloxy oxygen scavenger system. The new results are now shown on Figure 2 and Supplementary Figure 3. A 1-D autocorrelation function was also used to process the dynamic trajectories obtained using the glucose + gloxy oxygen system (Supplementary Fig. 3g). The fitting of the autocorrelation curves to bi-exponential decays does not show any significant difference between the parameters using an unpaired t-test (Supplementary Fig. 3g).

4) The dwell time data in Fig. 1d look like they may be better fit to a two-step mechanism, because the first bin has a lower occupancy than the second bin. This possibility could be better differentiated with slightly finer binning of the data.

We thank the Reviewer for the suggestion. We have re-fitted the dwell time distribution in Fig. 1c and also dwell distributions in other figures (Fig. 3 and Supplementary Fig. 6) by slightly adjusting the bin.

5) It is unclear how the "rates" (rate constants?) shown in Figs. 1f, 2e, and 4d are determined or. The Methods indicate that $n > 20$ events was used for rates. It is unclear if events mean transitions or traces, and this number events is quite small for determination of a rate constant from a dwell time plot. Also, the source of the error bar is not stated. The authors need to show the dwell time plots and the fits to the data for the reader to judge the comparison.

We thank the Reviewer for the suggestion. The transition rates shown in Fig. 1f and old Fig. 2e (new Supplementary Figs. 3d,e) represent the derived transition rates output from the TDP software after HaMMY analysis (Sean A. McKinney, Chirlmin Joo, Taekjip Ha, *Biophysical Journal*, 2006, 91: 1941-1951). We detailed description of the analysis on page 23. The error bars represent the standard deviation (SD) of the fit from the TDP software as demonstrated in the figure legends (Fig. 1f and Supplementary Fig.3f). Here 'events' means 'traces'. We have modified this text to make it clearer (legends in Fig. 1f and Supplementary Fig.3f).

Figure 4d shows the summary of the dwell time of the formed **CS** complex. The error bars represent the SD of two replicates. To better organize the graphs in Fig. 4, we only show the summary of the synapsis times. The dwell time plots and corresponding fits were shown in the new Supplementary Figs. 6c,d (old Supplementary Fig. 4).

6) Fig. 2c show a survival probability plot but it is unclear how the data were evaluated to generate this plot. They provide a reference to another paper; however, they do not appear to have the same issue of too few transitions in cited reference. The authors should more clearly explain their analysis. Perhaps the authors could take longer movies to circumvent this problem, since the donor is in solution and photobleaching should be less of a problem.

For the **FS** complex, the synapsis time includes the entire duration time (interval between the beginning and ending time points) of each synapsis event as shown in Fig. 1c. The synapsis time was calculated based on the Cy3 signal, regardless of the transitions. This means the synapsis time of **FS** complex contains both the near-zero E_{FRET} portion and detectable E_{FRET} portion (Fig.1c).

For the dwell time plot, we choose the traces with the entire synapsis process within the detection time window (~2 min), which means only the traces with synapsis beginning time and ending time points both within the 2 min time window were chosen for the dwell time plot (Fig. 1c). In the initial submission, for the Ku, X4L4, and DNA-

PKCs condition using the PCA + PCD oxygen scavenger system (old Fig. 2c), we did not get a sufficient number of this kind of trace (described above) for a dwell time plot. Therefore, all of the traces without a synapsis duration lasting for the entire movie period (2 min) were selected for the survival plot. The traces used for the survival plot include the synapsis either starting at the beginning of the movie (synaptic complex formed before the movie), or lasting to the end of the movie (synaptic complex still exists after the movie). We have added a detailed description of the survival plots in the Methods.

Because the PCA + PCD system blocks the kinase activity of DNA-PKCs, we repeated all the experiments involving DNA-PKCs using the buffer with the glucose + gloxy oxygen scavenger system. We obtained sufficient data for the dwell distribution fitting. We put the dwell time distribution and corresponding exponential fit in the new Fig. 2c. The error shown on the graph represents the SD of three independent replicates.

7) The discussion of synapsis efficiency on page 6 is hard to follow. The word synapsis appears to have different meaning throughout the paper (those that ligate, those that come together with any FRET, only those that come together with high FRET), which is confusing.

We thank the Reviewer for the suggestion. We have modified the text accordingly to make the terminology clearer. Synapsis in this manuscript means the association of the two separated dsDNAs (one immobilized on the slide, the other in the solution) mediated by NHEJ proteins. Since no 5'-PO₄ exists at any of the dsDNA ends, the two dsDNAs cannot be covalently ligated by X4L4, and the synapsis process is reversible in our system. Therefore, synapsis in this manuscript includes any association of the two dsDNA, which was monitored by the FRET signal in real-time. We found two primary kinds of synapsis – flexible synapsis (**FS**) and close synapsis (**CS**). The two dsDNA within **FS** complex are laterally aligned. The E_{FRET} of **FS** complex exhibits a broad distribution and has a main peak with a smaller E_{FRET} value. The two dsDNA within the **FS** complex could not be directly ligated (even with a 5'-PO₄) because of the lateral alignment. Within the **CS** complex, the two dsDNA ends are in an end-to-end configuration. The two dsDNA within **CS** complex are positioned to be ligated. The E_{FRET} distribution of the **CS** complex exhibits a single high FRET peak, whose FRET value is similar to that from the pre-ligated positive control. Therefore, the synapsis in this manuscript indeed includes low FRET (**FS**) and high FRET (**CS**) complexes.

None of the synapsis events in this manuscript can involve ligation. As suggested by the Reviewer, we carefully checked our manuscript and modified the text to make the description clearer.

8) On page 6, the authors state “A higher synapsis efficiency is observed for the end labeled configuration, suggesting that the binding of the two duplexes occurs preferentially at the DNA ends.” This result does not make sense. The complexes should be the same, unless the dyes are affecting with complex formation. How is efficiency calculated? What is the error?

We thank the Reviewer for this comment. We apologize for a mistake here about the description of the synapsis efficiency for the mid-point labeled DNA probe. The Reviewer is right in that the synapsis efficiency should be the same for the two cases no matter where the dye is located on the dsDNA. Here we should use “detected FRET pair counts/number (or normalized FRET pair counts/number)” instead of synapsis efficiency. We have modified our text and the Supplementary Figs. 1g and 4g accordingly.

As for the synapsis efficiency calculation, we show the normalized synapsis efficiency in this manuscript for the following reason. Since variances exist between the slides, and we repeat most of the reactions on different slides, we usually did a control (specified on the graphs, usually Ku plus X4L4 reaction condition) in the first chamber on each slide. The normalized synapsis efficiency was then obtained by normalizing the detected FRET pair number of each reaction to the FRET pair number of the control reaction on the same slide. The normalized synapsis efficiency of the control reaction is 1. Therefore, we did not show the error bar for the control reaction. We have a detailed description about the synapsis efficiency calculation in the Methods. The error bar represents the SD of at least two independent replicates for all the synapsis efficiency graphs. We have carefully checked all of the figure legends to make sure the statements about error are present in the legends.

Reviewers' comments:

Reviewer #1 (Remarks to the Author):

The manuscript has improved considerably as a result of authors' effort to revise it. This study is a direct rebuttal of data from multiple other labs using different systems with the main point of contention being the role of DNA-PKcs in DNA end synopsis. I am convinced that this protein is not required for synopsis in the authors' single molecule FRET system using the set of substrates and protein combinations tested here, but the broader point that DNA-PKcs is involved in end synopsis or not in vivo remains an open question and should be addressed in the Discussion.

Specific points:

1. The writing is really dense. I would encourage the authors to edit the text liberally to ease the readability of the paper and help the reader follow the logic of the experiments. The abbreviations for the different types of end synopsis are a bit confusing.
2. I asked the authors to show the gel in Figure 2a, but they have not added it to the manuscript.
3. The authors investigated the activity of DNA-PKcs and made changes to their buffers to ensure that the kinase activity is active. It is possible that DNA-PKcs works in concert with other NHEJ proteins not present in the authors' system to promote synopsis. Discuss this point please.
4. I asked the authors whether titrating down the concentration of X4L4 might reveal a requirement for DNA-PKcs, but they have not done this experiment.
5. I suggested that the authors test a few key compatible and incompatible overhang structures for comparison to blunt ends. As the authors state in the rebuttal, blunt ends are a good model for the most difficult synopsis events, but they represent only a small fraction of DSB ends in vivo. This should be discussed at least.

Reviewer #2 (Remarks to the Author):

The authors have done an impressive work to address the concerns of the reviewers. Prompt publication of the manuscript is recommended.

Reviewer #3 (Remarks to the Author):

This revised version has improved; however, there remain significant issues with the analysis (discussed below). Their data strongly support their conclusion that XLF promotes "close synopsis"; however, their data do not support their statement in the abstract that "The stoichiometry of XLF and its promotion of close synopsis indicate a role that is independent of a filament structure, with action focused at the very ends of each duplex". Their data do not provide any information about the stoichiometry of any of the proteins in the reaction. Their XLF concentration dependent data in Supp. Fig. 5 appear to show a sigmoidal dependence on the fraction of CS as a function of XLF concentration (note similar fraction at 16.7 and 50 nM XLF followed by significant increase), which would suggest 2 or more XLF proteins binding. Later in the paper, they state that "XLF functions in substoichiometric amounts"; however, although XLF may be in concentrations less than other proteins, it is in vast excess over DNAs attached to the surface (likely tens of pM) and only a subset of the DNAs exhibits a CS state, suggesting only a subset of the complexes contain XLF. Notably, the only evidence for the presence of any given protein in the complex is based on a change in E-FRET. Consequently, the simplest interpretation of a lack in change in E-FRET is that the protein is not interacting. Finally, it may be possible that the relatively short lengths of DNA used in the current study could contribute to the apparent lack of role DNA-PKcs.

Serious issues remain with the data analysis, especially the analysis using HaMMY, which can be highly dependent on user defined parameters. The problem becomes clear in comparing the results from HaMMY with the FRET efficiency distributions (Fig 1d & 1f). HaMMY produces two states: E→FRET 0 and 0.3 with the similar lifetimes (rates). These values and rates would in turn lead to the prediction of two states in the E→FRET distributions with one centered on 0 and the other centered on 0.3; however, the experimental distribution shows two peaks: a dominant peak at 0.09 and a smaller peak at ~0.3. The HaMMY analysis appears to be picking up transitions that may be noise, and it is also perhaps missing state(s) (i.e. 0.09). Either way the HaMMY analysis is not consistent with the experimental data shown in either Fig. 1 or Supp. Fig. 3. The same problem appears in the data in Supp. Fig. 2b and Supp. Fig. 3e & 3f, where HaMMY identifies 2 states (0 and 0.3) with rates that suggest ~2:1 population, but these states are not seen in the experimental data (Fig. 2b).

Another concern is that the data in Supp Fig 5 in the absence of XFL, appear to be significantly different (2 peaks of equal height) from those shown in Fig 1d, which appears to be under the same conditions.

I have uploaded an annotated version with comments that the authors may (or may not) find useful in their revisions.

Responses to Reviewer Comments (author responses in BLUE)

The authors would like to thank the Reviewers and the Editor for their time in evaluating our study and for making useful suggestions. We believe that we have addressed all of the comments below.

Reviewer #1

The manuscript has improved considerably as a result of authors' effort to revise it. This study is a direct rebuttal of data from multiple other labs using different systems with the main point of contention being the role of DNA-PKcs in DNA end synapsis. I am convinced that this protein is not required for synapsis in the authors' single molecule FRET system using the set of substrates and protein combinations tested here, but the broader point that DNA-PKcs is involved in end synapsis or not *in vivo* remains an open question and should be addressed in the Discussion.

We thank the Reviewer for agreeing with our findings. We have added sentences to the Discussion to acknowledge possible *in vivo* roles. No one yet knows the full range of roles that DNA-PKcs may play in end synapsis *in vivo*. We have discussed this issue on page 14 as the Reviewer suggested. We will test this possibility in cells in the future. As we note, for signal joint formation in V(D)J recombination, there is nearly universal agreement that DNA-PKcs is not required *in vivo*, based on many *in vivo* assays.

Specific points:

1. The writing is really dense. I would encourage the authors to edit the text liberally to ease the readability of the paper and help the reader follow the logic of the experiments. The abbreviations for the different types of end synapsis are a bit confusing.

We thank the Reviewer for the suggestion. We have modified the text to make it clearer and more readable.

2. I asked the authors to show the gel in Figure 2a, but they have not added it to the manuscript. We apologize for not mentioning this issue in the first Responses Letter. We indeed had included the gel results corresponding to Figure 2a in the manuscript. They were shown on Figure 6a (lane 7 and lane 9). Ku and X4L4 with or without DNA-PKcs in the system only mediate **FS**. The two dsDNAs within the **FS** complex are primarily laterally aligned, and therefore, they are not ready for covalent ligation. Therefore, as expected, no detectable ligation products corresponding to the reactions in Figure 2a were observed on the gel (Fig. 6a, lane 7 and lane 9).

3. The authors investigated the activity of DNA-PKcs and made changes to their buffers to ensure that the kinase activity is active. It is possible that DNA-PKcs works in concert with other NHEJ proteins not present in the authors' system to promote synapsis. Discuss this point please. We thank the Reviewer for the suggestion. We have added Discussion on page 14 to address this possibility as the Reviewer requested. We did include all of the core components and most of the ancillary components of NHEJ.

4. I asked the authors whether titrating down the concentration of X4L4 might reveal a requirement for DNA-PKcs, but they have not done this experiment.

We thank the Reviewer for the suggestion. We have done this experiment as the Reviewer requested. The data is now shown on Supplementary Figure 3d. As we expected in the first Response Letter, less synapsis was observed when using less X4L4 in the solution, and DNA-PKcs

does not have a substantial effect on the synapsis efficiency (Supp. Fig. 3d). The results further indicate that X4L4 plays a critical role for efficient end synapsis, and DNA-PKcs is not required.

5. I suggested that the authors test a few key compatible and incompatible overhang structures for comparison to blunt ends. As the authors state in the rebuttal, blunt ends are a good model for the most difficult synapsis events, but they represent only a small fraction of DSB ends in vivo. This should be discussed at least.

We thank the Reviewer for the suggestion. In the revised manuscript, we indeed had included brief Discussion for the overhang structures (line 428 – line 430, page 15; and line 497 – 498, page 17). Moreover, we have published some relevant work about overhang structures (PMID: 25941401 and PMID: 28930678). Further work will provide more detailed information about the compatible and incompatible overhang structures, but since there are thousands of possible overhang variations (sequence and length), such work will be a very long-term effort.

Reviewer #2

The authors have done an impressive work to address the concerns of the reviewers. Prompt publication of the manuscript is recommended.

We thank the Reviewer for recommending publication of our manuscript.

Reviewer #3

This revised version has improved; however, there remain significant issues with the analysis (discussed below).

Their data strongly support their conclusion that XLF promotes “close synapsis”; however, their data do not support their statement in the abstract that “The stoichiometry of XLF and its promotion of close synapsis indicate a role that is independent of a filament structure, with action focused at the very ends of each duplex”.

Their data do not provide any information about the stoichiometry of any of the proteins in the reaction.

We understand the Reviewer’s concerns. We have reworded our statements based on the Reviewer’s comments in the text. We have deleted the stoichiometry statement in the Abstract. XLF was reported to form a filament with XRCC4 on dsDNA, which was reported to bridge two dsDNA (PMID: 27437582). It was reported the XLF-XRCC4-dsDNA filament could slide along each other within the synaptic complex (PMID: 27437582). The interactions of the filaments are mainly within the middle portions of the dsDNA (PMID: 27437582). Here we found the main role of XLF is to promote the formation of a **CS** complex, within which the two ends of the dsDNA are in an in-line close contact at the ends and are ready for covalent ligation. Therefore, we stated “its promotion of close synapsis indicates a role that is independent of a filament structure, with action focused at the very ends of each duplex”.

Their XLF concentration dependent data in Supp. Fig. 5 appear to show a sigmoidal dependence on the fraction of CS as a function of XLF concentration (note similar fraction at 16.7 and 50 nM XLF followed by significant increase), which would suggest 2 or more XLF proteins binding. Later in the paper, they state that “XLF functions in substoichiometric amounts”; however, although XLF may be in concentrations less than other proteins, it is in vast excess over DNAs attached to the surface (likely tens of pM) and only a subset of the DNAs exhibits a CS state, suggesting only a subset of the complexes contain XLF.

We have reworded our statements based on the Reviewer’s comments. The summarized fraction of **CS** complex formed at different concentrations of XLF is shown in Fig. 3f. The Reviewer is correct

in stating that there is an increase of **CS** complex when using higher amounts of XLF. Based on the known interactions of these NHEJ proteins and the XLF concentration-dependent fraction of **CS** complex here, we suggest up to three XLF dimers at the dsDNA ends would further stabilize the **CS** complex (Discussion, page 15), which leads to accumulation of the **CS** complex.

Notably, the only evidence for the presence of any given protein in the complex is based on a change in E-FRET. Consequently, the simplest interpretation of a lack in change in E-FRET is that the protein is not interacting.

We have reworded our statements based on the Reviewer's comments in the text.

Finally, it may be possible that the relatively short lengths of DNA used in the current study could contribute to the apparent lack of role DNA-PKcs.

We thank the Reviewer for the suggestion. DNA-PKcs can bind to 18bp dsDNA and the binding to the 18 bp dsDNA can stimulate its kinase activity even without Ku (PMID: 9305651). When dsDNA is longer than 26 bp, Ku and DNA-PKcs form a productive complex which can further stimulates the kinase activity of DNA-PKcs comparing to that without Ku involved (PMID: 9742108). The shortest dsDNA used here is 74 bp, which is long enough for the binding of the core NHEJ proteins. Moreover, DSBs *in vivo* are unlikely to have a kb of naked DNA exposed.

Serious issues remain with the data analysis, especially the analysis using HaMMMy, which can be highly dependent on user defined parameters. The problem becomes clear in comparing the results from HaMMMy with the FRET efficiency distributions (Fig 1d & 1f). HaMMMy produces two states: E-FRET 0 and 0.3 with the similar lifetimes (rates). These values and rates would in turn lead to the prediction of two states in the E-FRET distributions with one centered on 0 and the other centered on 0.3; however, the experimental distribution shows two peaks: a dominant peak at 0.09 and a smaller peak at ~0.3. The HaMMMy analysis appears to be picking up transitions that may be noise, and it is also perhaps missing state(s) (i.e. 0.09). Either way the HaMMMy analysis is not consistent with the experimental data shown in either Fig. 1 or Supp. Fig. 3. The same problem appears in the data in Supp. Fig. 2b and Supp. Fig. 3e & 3f, where HaMMMy identifies 2 states (0 and 0.3) with rates that suggest ~2:1 population, but these states are not seen in the experimental data (Fig. 2b).

We understand the Reviewer's concerns. Here we mainly want to show the flexibility of the two dsDNA in the **FS** complex formed by Ku and X4L4. We realize the dynamic FRET trajectories of **FS** complex could suggest the flexibility of the two ends of dsDNA, and the transition rates between different states are not necessary and important here. Therefore, we have removed the HaMMMy analysis from the paper to simplify the manuscript.

Another concern is that the data in Supp Fig 5 in the absence of XFL, appear to be significantly different (2 peaks of equal height) from those shown in Fig 1d, which appears to be under the same conditions.

We understand the Reviewer's concern here. They were conducted under the same condition, but on different dates. The differences might be because of the variations of the experiments, which may be because of the relative activity of proteins used, and the slight differences of ambient temperatures and the slides. For reactions with Ku and X4L4, more than 95% synaptic complexes are **FS** complex. The **FS** complexes include numerous intermediate structures. The relative amount of different intermediate structures of the **FS** complexes may have a range of variation between experiments, which causes the differences of the FRET distributions of **FS** complexes. It is difficult to calculate the accurate ratios of different intermediate structures of **FS** complexes merely based on the FRET method. Therefore, we simply classify the synaptic complexes into the **FS** and **CS** complexes in the manuscript. Despite this variance for the **FS** complex, the ratios of **FS**

complexes and **CS** complexes from different batches are similar (97% **FS** in Fig. 1d, 95% **FS** in Supp. Fig. 5 top).

I have uploaded an annotated version with comments that the authors may (or may not) find useful in their revisions.

We thank the Reviewer's comments in the text. We have adjusted our words in the text accordingly based on the Reviewer's comments.

Additional point-by-point responses to the tracking comments in the Word file (pdf) from Reviewer 3:

Page 6, Line 155 to line 166:

One point of evidence for the side-by-side alignment of the **FS** complex is the results from the mid-point labeling probe. If two dsDNA molecules within the **FS** complex are predominately in an end-to-end configuration, we would not detect any synaptic molecules using the mid-point labeling since FRET would not occur between Cy3 and Cy5 that are separated by at least 40 bp. As shown in Supp. Fig. 1g and 1h, we could indeed detect synaptic molecules for the mid-point labeling probe. This result indicates that one dsDNA end could contact at the mid-section of the other within the **FS** complex, supporting the idea that the two dsDNA are in parallel alignment.

We would also like to highlight that we do not observe a large change in the FRET distributions for the DNA probes with Cy5 either at the end or at the mid-point, consistent with a side-by-side parallel arrangement. Furthermore, upon close inspection of the FRET distributions we indeed observe a larger fraction of the population exhibiting higher FRET (≥ 0.6) for the mid-point labeled probe compared to that for the end labeled probe (Fig. 1d and Supp. Fig. 1h) which strongly argues against an end-to-end configuration for the **FS** complex.

Page 6, Line 176, and Page 7, Line 181 and Line 200:

We have removed the HaMMY analysis here.

Page 8, Line 213:

We have reworded our words based on the Reviewer's comments. Instead of using "slide", we use the term "move" here.

Page 8, Line 215:

The dynamic trajectory of formed synaptic complex suggests the lateral alignment of the two dsDNA. The dynamics is not because of the motion of the two ends separated by proteins (end-to-end configuration), but because of the motion of the two dsDNA in parallel configuration for the following reasons. First, the synopsis efficiency is highly dependent on the X4L4 concentration (new Supp. Fig. 3d), suggesting X4L4 is critical for the synopsis even when DNA-PKcs is present in the solution. While Ku plus X4L4 mainly mediate the interactions at the mid-section of the dsDNA as discussed above. Second, if Ku, X4L4, and DNA-PKcs indeed can mediate end-to-end configured synaptic complexes (although we are skeptical about the statement), we would not observe any synaptic complexes merely from FRET signal based on previous study (PMID: 26990988). Because the end-to-end configured Ku-X4L4-DNA-PKcs-dsDNA synaptic complex exhibits no FRET (PMID: 26990988). Therefore, we think the two dsDNA are still laterally aligned within the detected synaptic complexes even when DNA-PKcs is present in the solution.

Page 8, Line 226:

We have removed all the HaMMY analysis from the traces.

Page 8, Line 236:

We have replaced the survival curves with dwell time graph (new Supp. Fig. 4b).

Page 9, Line 267:

We have reworded the term.

Page 10, Line 276:

We have explained the DNA length issue above.

Page 10, Line 299 to Line 301:

We have reworded the text.

Page 11, Line 312 to Line 314:

The main discrepancy from the Reviewer is about the sub-states of the **FS** complex. As we explained in the first Responses Letter, we observed two different kinds of synaptic complexes -- **FS** and **CS** complexes. The **CS** complex is relatively homogeneous. Within the **CS** complex, the two dsDNAs are aligned in end-to-end configuration (i.e. in-line close end-to-end contact) (Fig.6b), and are at the same distance as a ligatable nick control (as shown in Fig. 3d). Within the **FS** complex, the two dsDNAs are in parallel lateral alignment (i.e. side-by-side) (Fig. 6b). Therefore, one dsDNA end can contact any of multiple positions all along the length of the other dsDNA duplex within the **FS** complex. Within some of the **FS** complexes, the dsDNA can move along one another. The distance between the two dsDNA ends varies within different **FS** complexes, which results in a broad distribution of the E_{FRET} . Therefore, the **FS** complexes includes numerous intermediate structures. To simplify the description of the synopsis process in the manuscript, we focused on two primary states (represented by the two main peaks). The sub-states of the **FS** complex are still **FS** state.

The gel results in Fig.6a confirm our synopsis mode. The gel results show that XLF (Fig. 6a and Supplementary Fig. 9, lane 10) and/or PAXX (Fig. 6a and Supplementary Fig. 9, lane 11 and lane 12), which promote the formation of **CS** complex, can stimulate the blunt end ligation mediated by Ku plus X4L4 in our ensemble ligation assay. In contrast, Ku plus X4L4 (Fig. 6a and Supplementary Fig. 9, lane 9) or Ku, DNA-PKcs, and X4L4 (Fig. 6a and Supplementary Fig. 9, lane 7), which can only promote **FS**, cannot mediate detectable ligation of blunt ends. The gel results suggest that the two dsDNA ends within the **CS** complex are readily ligated because they are aligned in an appropriate end-to-end orientation and are in close contact, while the ends within the **FS** complex could not be covalently ligated because of the sub-optimal lateral side-by-side orientation.

Page 11, Line 319:

XLF and PAXX are paralogs. Here we simply want to say they both can promote **CS** complex formation through similar interactions with Ku and/or X4L4. They are not competitive with one another. Actually, one recent study shows that PAXX binds to Ku70 of Ku70/Ku80 heterodimer, and XLF binds to Ku80 (PMID: 30291363). It is reported that XLF can bind to XRCC4. But there is no known interaction between PAXX and XRCC4.

Page 13, Line 370:

The results from the mid-point labeling probe indicate the parallel side-by-side alignment of the two duplexes. Moreover, as we stated in the Discussion, once the ends can transiently pair via microhomology at the ends (instead of the case for blunt ends), Ku and X4L4 are sufficient to promote **CS** complex formation without XLF. This also suggests that the low FRET distribution of **FS** complex formed by Ku plus X4L4 for the blunt end dsDNA here is not because of the separation

of the two ends by Ku and/or X4L4 at the dsDNA ends as the Reviewer proposed in the text, but because of the side-by-side alignment of the two blunt dsDNA. This is because if Ku and/or X4L4 could separate the two ends of dsDNA, they could also separate the ends of two overhang dsDNAs. Therefore, we would not observe the **CS** complex formation for the overhang dsDNAs, but we do. We speculate that any factors that could facilitate the end-to-end configuration could stimulate the formation of **CS** complex. For the blunt end dsDNA, Ku and X4L4 can only promote the side-by-side **FS** when XLF and/or PAXX are absent.

Page 13, Line 393:

It is known that the ligation efficiency of blunt end dsDNA is much lower (PMID: 27703001; PMID: 27705800).

Page 14, Line 415:

We have explained the DNA length issue above.

Page 15, Line 455:

The stability of **CS** complex mentioned here means the stability of the synaptic structure, but not the longevity of the synaptic complex. The two ends are in an in-line close contact within the **CS** complex, and the dynamics of this state is limited. The two ends within the **CS** complex are readily ligated. Therefore, we state that the formation of **CS** state facilitates end protection and ligation.

Page 16, Line 461, and Page 17, Line 492:

We have deleted all the statements about the sub-stoichiometry of XLF.

Page 17, Line 496:

We have reworded the term.

Page 28, Line 896:

We agree that there are multiple intermediate structures of the **FS** complexes. But as we discussed above, the highest peak is used to represent all of the **FS** state to simplify the description of the synaptic complexes.

Page 33:

We have removed all of the HaMMY analysis from the traces.

REVIEWERS' COMMENTS:

Reviewer #1 (Remarks to the Author):

I am impressed with authors' effort to revise a very interesting, technically sound study.

The second revised paper is suited to publication in Nature Communications. I am confident that the study will stimulate lively discussions and research activities regarding end synopsis in NHEJ.

REVIEWERS' COMMENTS:

Reviewer #1 (Remarks to the Author):

I am impressed with authors' effort to revise a very interesting, technically sound study.

The second revised paper is suited to publication in Nature Communications. I am confident that the study will stimulate lively discussions and research activities regarding end synapsis in NHEJ.

We thank the Reviewer for supporting our research.